# Changes in the flowing drainage network and stream chemistry during rainfall events for two pre-Alpine catchments

Izabela Bujak-Ozga[1,2], Jana von Freyberg[1,2], Margaret Zimmer[3], Andrea Rinaldo[1,4], Paolo Benettin[1,5], and Ilja van Meerveld[6]

[1]EPF Lausanne, School of Architecture, Civil and Environmental Engineering, Lausanne, Switzerland
[2]Swiss Federal Institute for Forest, Snow and Landscape Research (WSL), Mountain Hydrology and Mass Movements, Birmensdorf, Switzerland
[3]University of Wisconsin, Madison, Department of Soil and Environmental Sciences, Madison, Wisconsin, USA
[4]Università di Padova, Department of Civil, Environmental and Architectural Engineering (DICEA), Padova, Italy
[5]University of Lausanne, Department of Earth Surface Dynamics, Lausanne, Switzerland
[6]University of Zurich, Department of Geography, Zurich, Switzerland

*Correspondence to*: Izabela Bujak-Ozga (hydrology@izabelabujak.com)

**Abstract.** Many headwater catchments embed non-perennial streams that flow only during wet conditions or in response to rainfall events. The onset and cessation of flow result in a dynamic stream network that periodically expands and contracts. The onset of flow can flush sediment and nutrients from previously dry streambeds and enhance the rates of carbon and nitrogen mineralization. The expansion of the flowing drainage network also increases hydrologic connectivity between hillslopes and streams because it decreases travel distances to the stream. However, datasets on the dynamics of the flowing drainage network and short-term changes in stream chemistry during rainfall events are rare. This limits our interpretation of hydrological processes and changes in stream chemistry during events.

Here, we present hourly measurements of solute concentrations and stable isotopes from precipitation and streamflow at the outlets of two 5-ha catchments in the Swiss pre-Alps during seven rainfall-runoff events in the snow-free season of 2021. Samples were also collected from soil- and groundwater across the catchments. We combine these data with 10-min information on the flowing drainage network length to infer the dominant runoff-generating mechanisms for the two experimental catchments.

Despite their proximity and similar size, soil and bedrock characteristics, the flowing drainage network dynamics were very different for the two catchments. In the flatter catchment (average slope: 15°), the stream network was more dynamic and expanded rapidly, up to 10-fold, while in the steeper catchment (average slope: 24°), it remained relatively stable (only a 2-fold change). The event water contributions were higher for the flatter catchment. The dilution of calcium at the time of the rapid expansion of the network and increase in discharge suggested that the contribution of rainfall falling directly on the stream channels is important, especially for the smaller events during dry conditions. During wet conditions, event water must have been delivered from areas outside the channels. In the flatter catchment with the more dynamic stream network, a "flush"

of nitrate was detectable, possibly due to the transport of material from previously dry stream segments. In the catchment characterized by a more stable flowing drainage network, such a flush was not observed and nitrate concentrations decreased, suggesting larger contributions from riparian groundwater with reducing conditions during rainfall events. Our experimental study not only highlights the large differences in stream network dynamics and stream chemical responses for neighboring catchments but also shows the value of fine-scale observations on both the channel network dynamics and stream chemistry to understand runoff generation mechanisms.

## 1 Introduction

More than half of the stream network globally consists of streams that cease to flow for a certain time of the year (Messager et al., 2021). This portion is expected to increase because of human-induced changes and climate change (Jaeger et al., 2014; Ward et al., 2020). The intermittent (i.e., non-perennial) stream network is a very dynamic system (Gregory and Walling, 1968) that often hosts high biodiversity and serves as a habitat to endemic species (Meyer et al., 2007; Acuña et al., 2014; Doods, 2004; Stubbington et al., 2017). Intermittent streams are also connectivity corridors (Rinaldo et al., 2018) that have important implications for metapopulation (Mari et al., 2014; Giezendanner et al., 2021) and ecosystem dynamics (Datry et al., 2023). However, major challenges persist in our understanding of how, when, and where flow in intermittent streams occurs (Fovet et al., 2021). Specifically, the link between runoff-generation mechanism, stream network expansion and contraction and biogeochemical processes has not yet widely been investigated (Covino, 2017; Zimmer et al., 2022).

The effects of flow intermittence and related drying and re-wetting cycles on nutrient cycling have been studied for individual stream reaches (Datry et al., 2023). The onset of flow can lead to the flushing of sediment (von Schiller et al., 2017) and the mobilization of coarse particulate organic matter (Lamberti et al., 2017) that were stored in the dry streambed. Temporarily connected streams can be significant sources of carbon (Thoms 2003; Thoms et al. 2005), organic matter and nutrients (Shumilova et al., 2019) to downstream rivers. The onset of flow in previously dry streambeds also increases biogeochemical processing rates (von Schiller et al., 2017, Burrows et al. 2017; Addy et al., 2019). The $CO_2$ efflux from dry streambeds can be substantial (Keller et al., 2020), compared to that from surrounding soils (Arce et al., 2019), and higher than from channels with flowing or standing water (von Schiller et al., 2014).

The dynamic variations in flow conditions along drainage networks can influence the quantity and quality of streamwater in downstream reaches (Brinkerhoff et al., 2024; Alexander et al., 2007). In addition to the changes in stream chemistry due to in-channel processes, stream chemistry is also affected by changes in the relative contributions of different flow pathways (e.g., Knapp et al., 2022). Water in intermittent streams may have a different chemical composition from that in perennial streams, e.g., because it contains less deep groundwater and is primarily fed by near-surface flow pathways. Warix et al. (2023) used CFC-11 and $^3$H measurements in intermittent streams and showed that deep flowpaths contributed to all streams, but that

cessation of near-surface flowpaths was responsible for stream drying. When the flowing drainage network expands, hillslope flow pathways become shorter (van Meerveld et al., 2019), which changes the hydrological connectivity between hillslopes and streams. This can also affect stream chemistry. Zimmer and McGlynn (2018) combined measurements of the wet portion of the drainage network with measurements of DOC concentrations in two 3.3 and 48.4 ha catchments in North Carolina, USA.

They found that DOC export and streamflow dynamics were driven by the connection and disconnection of lateral, longitudinal, and vertical source areas and associated changes in dominant flow pathways. Hale and Godsey (2019) showed that seasonal-scale DOC dynamics were strongly related to indices describing streamflow intermittency and that DOC concentrations were more stable in locations where the flow was more persistent. These studies demonstrate that flowing drainage network dynamics are closely linked to variations in streamwater chemistry. However, the relations are not yet well

established.

This lack of understanding of how the changes in flowing drainage network lengths affect solute concentration dynamics is caused by technical challenges in measuring both variables. Field surveys to determine the seasonal flowing/wet portions of the drainage network (e.g., Godsey and Kirchner, 2014; Botter and Durighetto, 2020; Durighetto et al., 2020; Warix et al.,

2021) and its dynamics during rainfall events (Durighetto and Botter, 2021; Bujak-Ozga et al., 2023)are time-consuming. Water presence sensors (Jaeger et al., 2012; Jensen et al., 2019; Kaplan et al., 2019; Zanetti et al., 2022) and multi-sensor monitoring systems (Assendelft and van Meerveld, 2019) are useful for determining the wet and flowing portion of the drainage network, respectively, but the number of measurement locations remains limited due to the time required for maintenance, especially in mountainous catchments where streambeds are unstable. Still, the number of available datasets from research

catchments is increasing. Recent research efforts have focused on describing the general patterns of the dynamics of drainage networks (Botter et al., 2021; Price et al., 2021; Durighetto and Botter, 2022), and the use of probabilistic (Durighetto et al., 2022) and machine-learning (Mimeau et al., 2024) methods and physically based models (Ward et al., 2018) to predict drainage network dynamics based on limited field data. Thus, high temporal resolution information on the dynamics of stream networks is now becoming available.


Considering the potential importance of intermittent stream dynamics on affect stream chemical responses during rainfall events, and the lack of high-resolution data for both the drainage network and solute concentrations, we investigated the dynamic changes in flowing drainage networks and streamwater chemistry in two Swiss pre-Alpine catchments. Specifically, we address the following research questions:

(1) How do solute concentrations and flowing drainage networks change during rainfall events, and how are these dynamics related to event characteristics?

(2) How do solute concentrations and flowing drainage network dynamics differ for two catchments with different geomorphic channel networks?

(3) What runoff-generation mechanisms are consistent with the observed solute concentrations and flowing drainage network

dynamics?

The joint measurements of the flowing drainage network and solute concentrations were taken during rainfall events in the snow-free season of 2021. The neighboring catchments have similar sizes, soils, and geology but differ in topography and the length of the channel network.

## 2 Methodology

### 2.1. Study sites

### 2.1.1 Location and topography

This study was carried out between June and October 2021 in two tributaries of the Erlenbach (Erl; 0.7 km2) research catchment in the Swiss pre-Alps (Stähli et al., 2021), located approximately 40 km southeast of Zurich (Fig. 1). The two studied tributaries are referred to as Lan and Cha, based on the local names of the areas that they drain (Langried and Chasperböden,

respectively). The 0.048 km$^2$ Lan catchment ranges from 1195 to 1292 m a.s.l. in elevation. With an average slope of 15°, Lan represents the flatter and lower part of the Erl catchment. The geomorphic channel network extends over most of the Lan catchment and is augmented by several ditches (Fig. 1). The Cha catchment has a similar size as Lan (0.048 km$^2$) but it is located at a higher elevation (1487 to 1656 m a.s.l) in the steeper (average slope 24°) part of the Erl catchment. The upper half of the Cha catchment is particularly steep, likely due to a fault zone between two types of Flysch bedrock (Fig.1; Bujak-Ozga

et al., 2023). In Cha, springs emerge near the bottom of the steepest slope and stream channels are located only in the lower, flatter portion of the catchment.

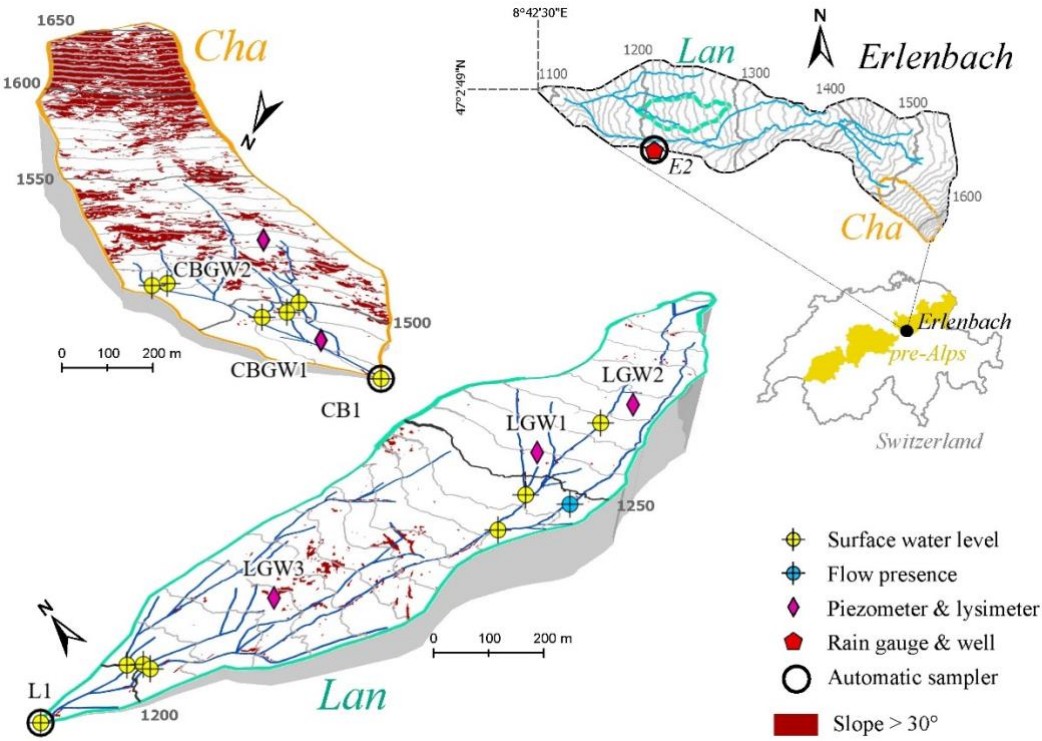

**Figure 1: Maps on the left show the two study catchments Langried (Lan; blue) and Chaspersböden (Cha; orange) and the monitoring network in each catchment. The maps on the right show the location of Cha and Lan in the Erlenbach (Erl) catchment (upper) and the location of the Erl catchment and the pre-Alps in Switzerland (lower). The grey lines represent the 5 m contour in Lan and Cha and the 10 m contour lines in Erl and are based on the digital elevation model with a 0.5 m resolution (SwissALTI3D, SwissTopo). The dark red shading represents areas with slopes >30°. The coordinate system is CH1903/LV03.**

### 2.1.2 Climate

The average annual precipitation at the Erlenhöhe climate station at 1210 m a.s.l. in the Erl catchment is 2266 mm (water years 1969–2019; Stähli et al. 2021), of which approximately one-third is snow. Although an up to 2 m thick continuous snow cover can be present from December to April, at lower elevations it is often interrupted by winter rainfall (von Freyberg et al. 2022). The mean monthly temperature ranges from -1.9 °C to 15.9 °C (von Freyberg et al. 2022). On average, it rains every second day in summer (van Meerveld et al., 2018) but the summer of 2021, and especially July 2021, were very wet. Total precipitation between June 1st and October 1st was 1154 mm, compared to an average between 2010 and 2020 of 954 mm (range: 609-1395 mm; Stähli, 2018). Total precipitation in July 2021 was 548 mm, almost three times the average of 190 mm and higher than measured between 2010 and 2020 (range: 7-530 mm; Stähli, 2018).

### 2.1.3 Geology, pedology, and land-use

The geology of the Erlenbach catchment is flysch (Hantke et al., 2022). The flysch bedrock is overlain by 1-2 meter thick clay-rich gleysols. Because the gleysols are locally shallow and have a very low permeability, groundwater levels remain close to the surface in a large part of the catchment, especially in the flatter areas (Rinderer et al., 2014). Holocene deposits interspersed by Pleistocene moraine (till) deposits can be found in the Lan catchment and the lower part of the Cha catchment (Hantke et al., 2022).

The Lan catchment is partly covered by forests, grasslands, and wet meadows. The coniferous forest is dominated by Norway spruce (*Picea abies*) and silver fir (*Abies alba*) (Stähli et al., 2021). In contrast, the Cha catchment is mainly covered by grassland and wet meadows, especially in the western part. There are a few isolated groups of trees, including Norway spruce (Picea abies) and silver fir (Abies alba), mainly in the eastern part of Cha (Stähli et al., 2021). The Cha catchment is used as a pasture for cattle during summer.

### 2.2. Hydrometric measurements and data

Stream water levels were monitored every 5 min with air pressure and temperature-compensated pressure transducers (CTD10, METER Group, Pullman, Washington, USA) and pressure transmitters (26Y, Keller AG, Winterthur, Switzerland) at ten locations in Lan and eight locations in Cha (Fig. 1). At the catchment outlets, stream water levels were measured behind a V-notch weir. These measurements were converted to discharge time series using the Kindsvater-Shen equation (Kulin and Compton, 1975). At all other locations, the pressure transmitters were placed directly in the middle of the channel.

For groundwater monitoring, 1.19-1.44 m deep piezometers were installed at three locations in Lan (LGW1-LGW3) and two locations in Cha (CBGW1, CBGW2). The piezometers were screened over the lowest 50 cm. Groundwater levels were monitored every 5 min using pressure transducer (LGW1) and transmitters (LGW2-3). For Lan, data are available from the 1[st] of June to the 27[th] of October 2021 for all measurement locations, except for LGW3 for which data are available from the 9[th] of September 2021 onwards. For Cha, data are available from the 18[th] of June to the 27[th] of October 2021 for all measurement locations, except for the outlet (CB1) for which data are available already from the 1[st] of June 2021. Data gaps of up to a few hours occurred sporadically due to data transmission errors and measurement disturbances (e.g., during sensor maintenance or water sample collection).

Precipitation data at the Erlenhöhe climate station near Lan (E2; Fig. 1) were provided by the WSL's Mountain Hydrology and Mass Movements research unit. These data are available at a 10 min resolution and were measured using a tipping bucket rain gauge (Pluvio2 L400 RH, Ott Hydromet GmbH, Switzerland).

## 2.3. Channel flow state surveys

On the 2$^{nd}$ and 4$^{th}$ of June 2021, we mapped the complete geomorphic channel network in both catchments. We defined a channel as a depression or landscape feature where directed surface flow occurs or where there are visible signs that flow occurred in the recent past (i.e., in the last few months). The entire channel network was divided into 85 and 48 reaches with a median length of 18 m and 13 m in Lan and Cha, respectively (cf. supplementary information to Bujak-Ozga et al. 2023). The start and end of the reaches were defined based on similar hydrologic conditions along each reach (i.e., expected drying and rewetting patterns based on locally observed water level, wetness, slope, channel width, and streambed material; Fig.S1).

We carried out eighteen spatial streamflow presence or absence mapping surveys in Lan and fifteen surveys in Cha. During each mapping survey, we classified all reaches as either "flowing" or "not flowing". These surveys took approximately 5 hours in Lan and 3 hours in Cha. To better delineate the flow regime within the catchment and help achieve higher reproducibility, we further categorized "flowing" reaches as weakly trickling (<1.0 [l min$^{-1}$]), trickling (2.0-1.0 [l min$^{-1}$]), weakly flowing (5.0–2.1 [l min$^{-1}$]), or flowing (>5.0 [l min$^{-1}$]) based on a visual estimate of the flow or bucket measurements. The "not flowing" reaches were further categorized into either standing water, wet streambed, or dry streambed. Besides the regular catchment-wide mapping surveys, we reported our visual observations of the channel flow state for selected reaches during equipment maintenance. We gathered the georeferenced measurements of the channel flow state, water levels, photos and videos using the TempAqua App iOS (Bujak-Ozga, 2023). At one location (marked with blue in Fig. 1) mapping surveys were complemented by continuous flow presence measurements with a multi-sensor monitoring system (Assendelft and van Meerveld, 2019). Because the flow needs to be higher than 1.0 [l min$^{-1}$] for the flow sensor (propeller) to register flow, we only used the data when the sensor recorded flowing water. We did not use the sensor measurements when it registered the not-flowing state because it could also be flowing at a rate between 0 and 1.0 [l min$^{-1}$].

## 2.4. Hydrochemical data

### 2.4.1 Sample collection

Between the 5$^{th}$ of August and the 22$^{nd}$ of October 2021, we collected water samples from the streams, soil, and groundwater in the Lan and Cha catchments and from rainfall. In total, we collected 149 and 136 streamflow samples at the outlet during events, 98 and 118 streamflow samples at the outlet during baseflow, 68 rainfall samples, 55 and 37 soil water samples, and 37 and 29 groundwater samples for the Lan and Cha catchments, respectively. Stream water samples were collected at the outlets of the Lan and Cha catchments hourly during seven and six rainfall events, respectively, using MAXX autosamplers (P6, MAXX GmbH, Germany), as well as shortly before events. In addition, we conducted a spatial sampling campaign during dry (baseflow) conditions on the 7$^{th}$ of September 2021. Grab samples were collected at the beginning of every channel reach

that had standing or flowing water (thirty samples in Cha and nine in Lan). Rainwater samples were collected hourly during the same events at location E2 (Fig.1) using a MAXX autosampler as well.

Groundwater samples and soil water samples were collected before and after rainfall events and during the spatial sampling campaign. Soil water samples were collected using suction lysimeters installed at a depth of 30-35 cm, i.e., at the boundary between the rooting zone and the denser clay of the gleysols. The applied pressure was ~60kPa. Groundwater samples were collected from the piezometers. The longest period without groundwater and lysimeter sample collection was nine days

(between the 18th and 27th of August).

### 2.4.2. Laboratory analysis

All water samples were filtered using 0.45 μm Teflon membrane filters and stored refrigerated before analysis at the WSL central laboratory in Birmensdorf, Switzerland. The samples were analyzed for major anions using ion chromatography

(ICS3000, Dionex Corporation, USA). For the analysis of major cations, the samples were acidified ($HNO_3$) and subsequently analyzed using optical emission spectrometry with inductively coupled plasma (ICP-OES Optima 7300, PerkinElmer, Inc., USA). Solute concentrations are reported as the full solute in mg $l^{-1}$, not as the elemental component. The samples were analyzed for the stable isotopes of hydrogen and oxygen of the water molecule using laser isotope spectrometry (IWA-45-ER, Los Gatos Research, ABB Ltd., Switzerland). The stable isotope values are reported relative to the V-SMOW2 standard using

the delta ($\delta^2H$ and $\delta^{18}O$) notation. The measurement accuracy is ±1.0‰ and ±0.5‰ for $\delta^2H$ and $\delta^{18}O$, respectively, while the measurement precision is ±2‰ and ±1‰ for $\delta^2H$ and $\delta^{18}O$, respectively.

### 2.5. Data analyses

### 2.5.1 Rainfall events

We divided the precipitation and discharge time series into rainfall-runoff events. We define the start of a rainfall-runoff event as the onset of precipitation resulting in a substantial increase (>150%) in discharge at the Lan outlet. The end of an event either marks the time when discharge returned to the pre-event level or the start of the following event. Based on this definition of events, there were twelve events. Hydrochemical data were collected for seven and six of them in Lan and Cha, respectively. The event starts and end times are the same for both catchments because the time difference in the streamflow response was

minimal due to the proximity of the two catchments and the samples were collected only at an hourly resolution.

We determined the antecedent wetness conditions for the twelve events based on the average discharge during the 48 hours before the event ($Q_{48}$). We chose $Q_{48}$ because it reflected the groundwater level measured in the Erl catchment (E2). We use a

$Q_{48}$ of 4.5 [l min$^{-1}$] as a threshold to divide the events into those that had dry antecedent conditions, and those with wet antecedent conditions (cf. Bujak-Ozga et al., 2023) because $Q_{48}$ was < 4.5 [l min$^{-1}$] when the groundwater levels had dropped to a relatively stable level of 142 cm below the surface at location E2 As a second indicator of the antecedent wetness conditions, we calculated the cumulative precipitation during the 48 hours before the start of the event ($P_{48}$; cf. Kiewiet et al., 2019). $Q_{48}$ and $P_{48}$ were correlated (Spearman rank correlation $r_s$: 0.44, p=0.023) and therefore, we mainly report $Q_{48}$ in the text.

### 2.5.2 Flowing drainage network and flow persistency

We obtained time series of the flowing drainage network lengths (FDNL) and spatial maps of the flow/no-flow conditions for each channel reach at a 10-min resolution using the CEASE method (Bujak-Ozga et al., 2023). In short, this method combines the data from the mapping surveys with the data from the water level sensors to obtain time series of "flow" and "no flow" for each reach. The water level at the times of the surveys is used to determine for each reach the threshold water level above which the stream reach was flowing. This threshold is then used to obtain a time series of "flow" and "no flow" for the reach. This is done for each of the water level sensors. The time series of "flow" and "no flow" for the different water level sensors are compared and a final decision on "flow" or "no flow" is based on the majority vote for each time step. This is repeated for each reach, leading to a continuous time series of "flow" and "no flow" for each stream reach and spatial maps of the flow/no-flow conditions for each channel reach at a 10-min resolution. We used these spatial maps, to visually assess the connectivity of the flowing stream network. By summing up the length of all flowing reaches for each time step, we obtain the FDNL for the entire catchment. Because the total length of the all the channels differs for both the Lan and Cha catchments, we normalized the FDNL values by dividing them by their maxima. Thus, fFDNL is the fraction of the total drainage network that (according to the CEASE method) had flowing water. Finally, for each channel reach we calculated the fraction of time it was flowing, which we refer to as the local flow persistency. Here a value of 1 indicates that according to the CEASE method the reach was always flowing and a value of 0 indicates that the reach was never flowing. Detailed analyses by Bujak-Ozga et al. (2023) showed that the method selected the right flow state for >94% of the time that visual observations of flow or no flow conditions were available.

### 2.5.3 Characterisation of groundwater chemistry

The hydrochemical composition of groundwater can be spatially very heterogeneous, as shown for the neighboring Studibach catchment by Kiewiet et al. (2019). To characterize the hydrochemical composition of groundwater, we used three different sets of water samples collected during baseflow conditions: (1) groundwater samples, (2) streamwater samples from the catchment outlets during baseflow conditions and (3) streamwater samples collected during the snapshot campaign. Here, we assume that streamwater samples collected during baseflow conditions reflect the average hydrochemical composition of the

groundwater that contributes to streamflow. The results from the snapshot campaign and the groundwater samples are useful to assess the spatial variability of the shallow groundwater before it reaches the catchment outlet.

### 2.5.4 Hydrograph separation and end-member mixing analysis

We calculated the event-water fractions ($f_e$) for the streamwater samples using the isotope data in a two-component hydrograph separation analysis, following the procedure described in von Freyberg et al. (2018). We use these event water fractions as a proxy for rain falling on the stream channels and saturated areas (direct rainfall runoff) or very fast flow of event water through macropores in the topsoil (depth 0-30 cm). For four events, isotope-based hydrograph separation was not possible because the $\delta^2H$ values for streamflow and precipitation partly overlapped (Fig.S3, Fig.S5 - S7, Table 1). We estimated the uncertainties

in $f_e$ using the Gaussian error-propagation method (Genereux, 1998). The oxygen and hydrogen isotopes data yielded similar results, thus, we report only the results based on hydrogen stable isotope data based on the better precision to measurement range ratio.

The calculated event water fractions were used together with the concentrations in rainfall and the pre-event water to determine

the "expected streamwater concentration" based on mixing of rainfall and pre-event water (i.e., the concentration that should be expected if streamflow consisted only of a mixture of pre-event water and event water). Deviations from this expected concentration are either due to the contributions of different sources or biogeochemical reactions.

Furthermore, we evaluated the flow contributions from the subsoil (depth of approximately 30 cm; $f_{sw}$) based on the End

Member Mixing Analysis (EMMA; Christophersen and Hooper, 1992) using the silica concentrations and hydrogen isotope data for the soil water, rainfall and baseflow (outlet) samples. Both isotope-based hydrograph separation and EMMA were used to better understand the contributions of different flow pathways to streamflow and thus surface and subsurface connectivity.

## 3 Results and discussion

### 3.1 Chemical composition of the different water sources

### 3.1.1 Rainfall

Solute concentrations in rainfall were very low and close to the detection limits for all measured solutes, except nitrate and sulfate whose median concentrations were 0.6 and 0.3 [mg l$^{-1}$], respectively (Fig. 3). The stable isotope composition in rainfall varied widely (median = -72.8‰ and IQR = 52.8‰ for $\delta^2H$; Fig. 2), especially between rainfall events (Fig. 4b, Fig. S4b-S8b).


| Category | Location | Ca | K | Na | SO4 | Cl | Mg | Si | PO4 | NO3 | Mn | Fe | $\delta^2H$ |
|---|---|---|---|---|---|---|---|---|---|---|---|---|---|
| Rainfall | E2 | 0.11 (0.05–0.2) | 0.1 (0.1–0.1) | 0.04 (0.04–0.17) | 0.31 (0.12–0.55) | 0.05 (0.02–0.15) | 0.02 (0.01–0.03) | 0.1 (0.1–0.1) | 0.07 (0.07–0.07) | 0.6 (0.28–1.17) | 0 (0–0) | 0.01 (0.01–0.01) | −72.8 (−91.34–−38.45) |
| Lan Streamflow | Lan spatial | 67.63 (67.28–69.51) | 1 (0.88–1.03) | 18.59 (16.07–18.89) | 22.7 (20.62–24.07) | 0.95 (0.91–1) | 7.16 (6.1–7.61) | 2.19 (1.91–2.35) | 0.07 (0.07–0.07) | 0.38 (0.27–0.42) | 0 (0–0) | 0.02 (0.01–0.03) | −71.7 (−72.89–−70.74) |
| Lan Streamflow | L1 baseflow | 63.57 (62.83–64.99) | 0.81 (0.73–0.89) | 13.65 (10.38–15.36) | 18.6 (13.78–19.84) | 0.95 (0.7–1.02) | 5.43 (4.8–5.87) | 1.65 (1.51–1.76) | 0.07 (0.07–0.07) | 0.4 (0.36–0.46) | 0 (0–0) | 0.04 (0.03–0.05) | −70.7 (−71.6–−69.17) |
| Lan Streamflow | L1 all samples | 47.16 (28.99–62.08) | 0.81 (0.7–1.04) | 3.19 (1.15–11.72) | 9.6 (4.81–17.03) | 0.64 (0.43–0.96) | 2.82 (1.52–4.79) | 0.96 (0.7–1.48) | 0.07 (0.07–0.07) | 0.77 (0.46–1.12) | 0 (0–0) | 0.07 (0.04–0.08) | −66.08 (−70.49–−62.02) |
| Lan Groundwater | LGW3 | 83.63 (77.48–85.3) | 1.43 (1.35–1.5) | 73.12 (59.02–95.38) | 105.44 (42.64–149.85) | 4.99 (3.5–5.98) | 13.47 (11.81–14.59) | 2.04 (1.58–2.13) | 0.07 (0.07–0.07) | 0.11 (0.1–0.11) | 0.46 (0.29–0.53) | 0.01 (0.01–0.01) | −74.6 (−76.35–−73.1) |
| Lan Groundwater | LGW2 | 85.78 (79.34–92.83) | 0.86 (0.58–1.11) | 33.25 (31.85–51.38) | 11.48 (10.49–15.24) | 0.62 (0.51–0.91) | 5.67 (5.26–6.13) | 2.05 (1.79–2.25) | 0.07 (0.07–0.07) | 0.07 (0.03–0.14) | 0.41 (0.33–0.48) | 0.01 (0.01–0.02) | −73.8 (−76.55–−71.89) |
| Lan Groundwater | LGW1 | 101.59 (98.24–108.18) | 0.33 (0.24–0.36) | 3.65 (3.24–4.09) | 0.44 (0.38–0.54) | 0.12 (0.1–0.23) | 4.48 (4.02–4.76) | 1.55 (1.44–1.62) | 0.07 (0.07–0.07) | 0.02 (0.02–0.05) | 0.57 (0.56–0.6) | 0.07 (0.02–0.53) | −78.98 (−79.43–−77.95) |
| Lan Soil water | LGW3 | 60.67 (49.59–68.68) | 0.62 (0.48–0.78) | 2.81 (2.65–2.96) | 9.16 (2.13–16.03) | 0.18 (0.14–0.21) | 39.48 (38.08–44.58) | 41.26 (39.27–43.06) | 0.12 (0.07–0.33) | 0.04 (0.02–0.06) | 0 (0–0) | 0.01 (0.01–0.01) | −77.1 (−78.35–−76.3) |
| Lan Soil water | LGW2 | 131.86 (122.49–140.38) | 0.3 (0.24–0.34) | 4.07 (3.89–4.23) | 0.16 (0.14–0.21) | 0.12 (0.09–0.17) | 22.92 (19.69–40.05) | 36.11 (33.84–47.94) | 0.07 (0.07–0.14) | 0.03 (0.02–0.05) | 1.22 (0.39–1.51) | 0.01 (0.01–0.03) | −74.14 (−75.05–−73.42) |
| Lan Soil water | LGW1 | 119.65 (104.61–154.39) | 0.25 (0.1–0.33) | 1.56 (1.5–1.64) | 0.25 (0.23–0.35) | 0.15 (0.1–0.22) | 34.6 (31.15–47.67) | 53.36 (47.31–60.14) | 0.53 (0.32–0.6) | 0.02 (0.02–0.03) | 0.88 (0.03–1.59) | 0.01 (0.01–0.03) | −78.64 (−80.3–−75.7) |
| Cha Streamflow | Cha spatial | 65.25 (63.71–67.97) | 0.38 (0.32–0.48) | 1.1 (1–2.24) | 5.91 (2.97–7.91) | 0.26 (0.2–0.37) | 2.82 (2.2–3.74) | 1.33 (1.22–1.47) | 0.07 (0.07–0.07) | 0.42 (0.09–0.5) | 0 (0–0.01) | 0.01 (0.01–0.01) | −75.35 (−76.9–−74.12) |
| Cha Streamflow | CB1 baseflow | 63.08 (61.69–65.03) | 0.62 (0.51–0.91) | 1.65 (1.56–1.74) | 6.92 (6.66–7.36) | 0.48 (0.39–0.6) | 3.1 (3.01–3.21) | 1.32 (1.3–1.34) | 0.07 (0.07–0.07) | 0.39 (0.31–0.42) | 0 (0–0.01) | 0.01 (0.01–0.01) | −74.1 (−74.7–−72.5) |
| Cha Streamflow | CB1 all samples | 56.6 (44.74–63.09) | 1.18 (0.73–1.65) | 1.34 (1–1.63) | 5.31 (3.38–6.68) | 0.58 (0.44–0.87) | 2.65 (1.98–3.1) | 1.25 (1.05–1.33) | 0.07 (0.07–0.07) | 0.29 (0.22–0.39) | 0 (0–0.01) | 0.04 (0.02–0.08) | −72.2 (−73.9–−70.1) |
| Cha Groundwater | CBGW2 | 20.13 (18.49–22.41) | 0.1 (0.1–0.1) | 0.79 (0.58–0.94) | 1.17 (0.9–1.46) | 0.32 (0.25–0.6) | 0.39 (0.36–0.42) | 1.32 (1.21–1.41) | 0.07 (0.07–0.07) | 0.17 (0.07–0.23) | 0 (0–0.01) | 0.02 (0.02–0.04) | −68.64 (−70.33–−67.75) |
| Cha Groundwater | CBGW1 | 32.95 (23.43–34.55) | 0.1 (0.1–0.1) | 1.52 (1.46–3.72) | 0.41 (0.3–0.69) | 0.16 (0.06–0.24) | 0.75 (0.71–0.8) | 1.63 (1.57–1.68) | 0.07 (0.07–0.07) | 0.02 (0.02–0.06) | 0.47 (0.17–0.51) | 0.33 (0.03–0.47) | −81.4 (−82.06–−79.35) |
| Cha Soil water | CBGW2 | 75.48 (72.8–86.81) | 0.2 (0.1–0.26) | 0.5 (0.48–0.58) | 0.67 (0.56–0.72) | 0.3 (0.15–0.37) | 48.87 (33.75–96.1) | 56.34 (52.11–63.69) | 0.84 (0.65–0.94) | 0.18 (0.1–0.23) | 0 (0–0) | 0.01 (0.01–0.01) | −64.72 (−65.67–−64.17) |
| Cha Soil water | CBGW1 | 61.19 (56.87–63.51) | 0.1 (0.1–0.2) | 1.18 (1.11–1.29) | 0.18 (0.15–0.21) | 0.17 (0.12–0.22) | 35.48 (33.65–40.88) | 53.12 (50.74–57.99) | 0.44 (0.38–0.53) | 0.02 (0.02–0.05) | 0 (0–0.02) | 0.01 (0.01–0.01) | −81.95 (−83.47–−78.7) |

**Figure 2: Median solute concentrations (and interquartile range IQR in parentheses) for all measured solutes and locations sampled in the Lan and Cha catchments during the entire monitoring period (including baseflow conditions and rainfall events). All solute concentrations are in [mg l⁻¹]. Note that the intensity of each colour reflects the median solute concentration. Location names are shown on the right (see Figure 1 for the map). The terms "Cha spatial" and "Lan spatial" refer to the samples collected during the spatial sampling campaign during low flow conditions on the 7th of September 2021 in the Cha and Lan catchments, respectively.**


### 3.1.2. Groundwater

*Comparison of the different datasets*

Solute concentrations in Lan's groundwater were spatially variable. They differed among the three locations in Lan (Fig. 2). Moving uphill from LGW1 to LGW3, calcium concentrations decreased, while potassium, sodium, sulfate, chloride, and magnesium concentrations increased. A high spatial variability in groundwater concentrations is not surprising as Kiewiet et al. (2020) showed for the neighboring Studibach catchment that calcium concentrations in shallow groundwater can vary by 22 [mg l$^{-1}$] (standard deviation). Walker et al. (2003) attributed a high spatial variability in the groundwater chemistry in
headwater catchments to short flowpath lengths and limited mixing within a catchment. This supports a marked dependence on local conditions unbuffered by transport phenomena. On the contrary, the chemical composition of groundwater was similar for the two monitoring locations in Cha (Fig. 2; Fig. S1-S2).

The groundwater concentrations in the Lan catchment were similar to those collected in baseflow at the outlet, except for
calcium and manganese (higher in groundwater), and nitrate, iron, and sulfate (lower in groundwater) (Fig. 3). In the Cha catchment, groundwater differed from baseflow at the outlet for most solutes (Fig. 3). The calcium concentrations in baseflow at Cha were similar to those in Lan but higher than measured in the groundwater. This suggests that the piezometers in Cha may not be fully representative of all the groundwater that contributes to baseflow and that another, perhaps deeper, source of groundwater contributes to streamflow at the outlet of Cha. This would be consistent with the statistically significant difference
between the concentrations in the groundwater and baseflow for the majority of other solutes as well (Fig. 3).

For both catchments, there was no statistical difference between the baseflow samples taken at the outlet and those taken throughout the catchment, except for potassium (both catchments) and chloride (in Cha only) (Fig.S2), but even for these solutes, the differences in the concentrations were very small (e.g. around 0.2 [mg l$^{-1}$] for potassium). The similarity of the two
types of baseflow samples suggests that the samples collected at the outlets are representative for the baseflow and we therefore assume that they represent the integrated groundwater signal.

*Description of baseflow chemistry*

Baseflow was in both catchments characterized by high concentrations of weathering-derived solutes. Furthermore, the
groundwater concentrations were much higher and more spatially variable in Lan than in Cha groundwater, except for sulfate (Fig. 3). The more acidic conditions in the forested Lan catchment could lead to a higher solubility and higher concentrations weathering derived solutes (Gal et al, 1996; Berthrong et al., 2009; Huang et al., 2022) in Lan than Cha.

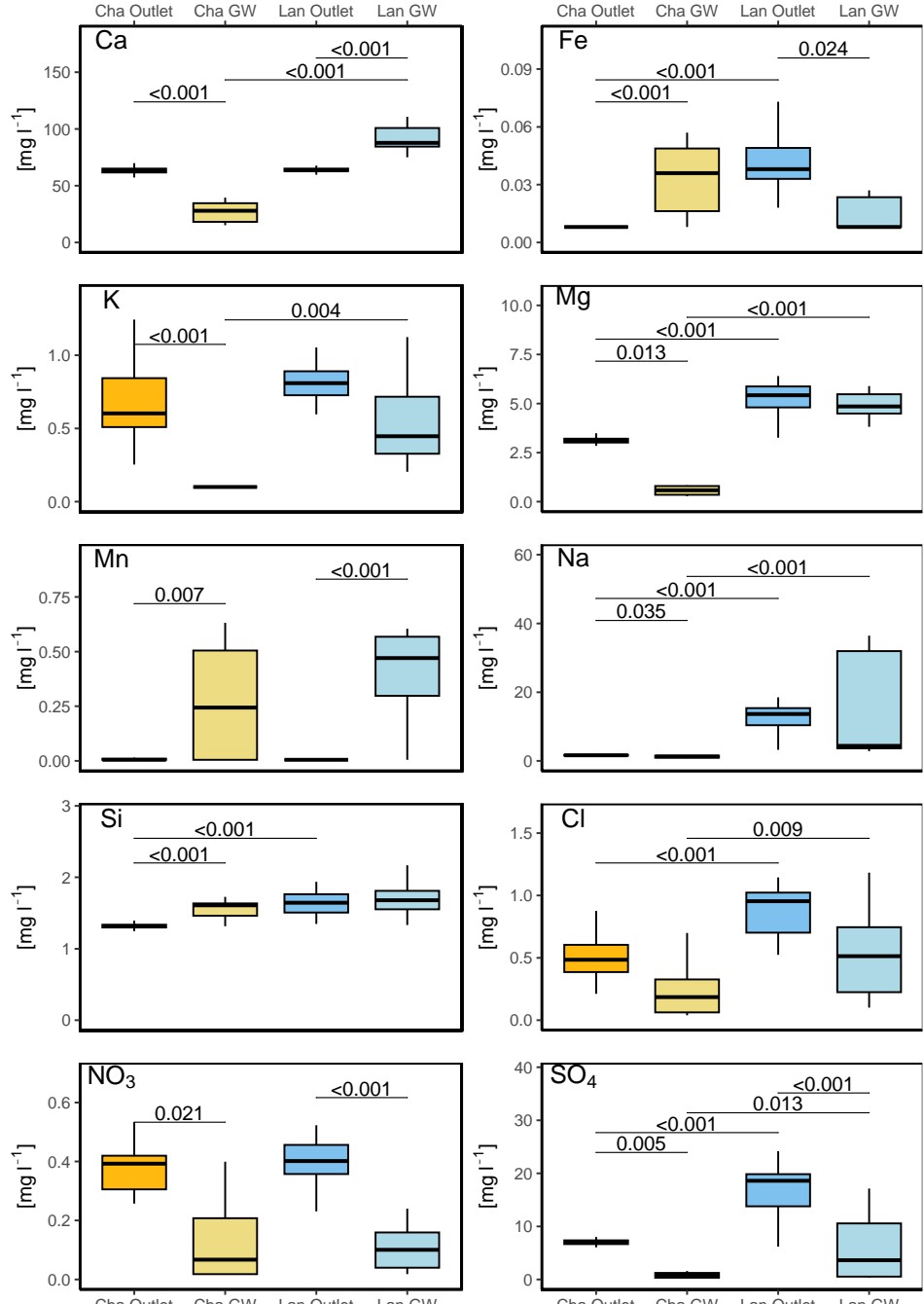


**Figure 3: Boxplots of the concentrations of selected solutes in baseflow at the catchment outlet (i.e., locations L1 and CB1) and groundwater (i.e., LGW1-LGW3, and CBGW1-2) in the Lan and Cha catchments. The boxes extend from the 1st to the 3rd quartile and the whiskers from the minimum to the maximum values. The lines represent the median values. The thin horizontal lines above the boxes indicate significant differences (p<0.05) in the median concentrations in baseflow and groundwater (intra-catchment**

**comparison) or between the two catchments (inter-catchment comparisons) according to the Kruskal-Wallis test with a Dunn post-hoc test.**

Calcium and sulfate were the most abundant solutes in both catchments. Calcium concentrations are high due to the weathering of the limestone and marl in the flysch bedrock. However, baseflow concentrations at the Lan outlet were lower and less variable than in the groundwater (as shown by other studies in this area, e.g., Fischer et al., 2015), suggesting that the oversaturated calcium in groundwater precipitates once it reaches the streams and physical conditions change.

Sulfate and sulfur concentrations in Lan groundwater were spatially and temporarily variable and generally higher than in Cha groundwater and baseflow (Fig. 2-3). Fischer et al. (2015) suggested that in Erlenbach (Erl) and five neighboring catchments Anhydrite-rich groundwater from deep aquifers could contribute to streamflow. It is possible that the two wells (LGW3, LGW2) located closer to the main channel that cut into the flysch bedrock, received water from a deeper and Anhydrite-rich source. Kiewiet et al. (2019) suggested that high sulfate concentrations in groundwater in the neighboring Studibach catchment could be related to pyrite weathering. However, in Lan and Cha, the locations with high sulfate concentrations were partially different from those with high iron concentrations. Therefore, the contribution of a deeper Anhydrite-rich source of water in the Erlenbach catchment explains the variability in our data better than pyrite weathering.

Chloride concentrations were less variable (and lower) in Lan than Cha (Fig. 2 - 3). Chloride concentrations in baseflow (outlet) were much higher (with medians of 0.95 and 0.48 [mg L$^{-1}$] for Lan and Cha, respectively) than in rainfall (median = 0.05 [mg L$^{-1}$]), which is usually considered the main source of chloride in undisturbed catchments (e.g., Peck and Hurle, 1973; Neal and Kirchner, 2000). Knapp et. al (2020) found higher than expected concentrations of chloride in the Erlenbach during the growing season but, when they considered the whole hydrologic year, the chloride concentrations could be reasonably well explained by the precipitation inputs. Schleppi et al. (1998) also reported relatively high chloride concentrations in Erlenbach stream water compared to the inputs and suggested that dry deposition may contribute to the higher chloride output.

Nitrate concentrations were very low in groundwater (median concentrations < 0.1 [mg l$^{-1}$]), but significantly higher in baseflow (Fig. 2). Slightly higher nitrate concentrations were only detected in one well, located on the Cha hillslope (median = 0.17 [mg l$^{-1}$] at location CBGW2; Fig. 3). Nitrate in the Erlenbach catchment originates from precipitation with minor inputs from grazing cattle in the upper catchment (Fig. 3; Knapp et. al, 2020).

### 3.1.3 Soil water

Soil water at a depth of ~30 cm was more enriched in silica, magnesium, phosphorus, and orthophosphate than groundwater (Fig. 3) but concentrations in nitrate, sulfur, sulfate, sodium, potassium, iron, and chloride were relatively low in soil water compared to the groundwater (Fig. 2). Similar to the groundwater, soil water concentrations were higher in Lan than Cha (Fig. 2). The temporal variations in soil water concentrations at a sampling site were smaller than the differences between the

sampling sites (Fig. 3). This high spatial variation compared to the temporal variation was also observed for the neighboring Studibach catchment (Kiewiet et al., 2019). High concentrations of magnesium in soil water were found in the neighboring Studibach catchment as well (Kiewiet et al, 2020). The generally low concentrations of silica and magnesium in baseflow suggest that soil water does not contribute substantially to baseflow and that streams were mainly fed by groundwater (Fig. 2-3). Previous research in an area close to Lan showed that nitrate was present in low concentrations only in the topsoil (0-10

cm depth). In deeper layers, nitrate, and ammonia concentrations were negligible, and dissolved organic nitrogen was the predominant form of nitrogen (Hagedorn et al., 2001).

### 3.2. How do the flowing drainage networks and stream chemistry change during events

### 3.2.1. Description of the events

The events on 2021-09-10, 2021-09-29, and 2021-10-21 were small (total precipitation between 11 and 16 mm) and had dry antecedent conditions (Table 1). However, the 2021-10-21 event was shorter than the other two events, resulting in a larger streamflow response and higher event water fractions compared to the other two small events. The events on 2021-10-12 (P = 23 mm) and 2021-08-28 (P = 28 mm) were moderate in size and occurred when antecedent conditions were wet. For Lan, this event resulted in a large streamflow increase (Table 1). This was not the case for Cha because part of the precipitation fell as

snow. The 2021-09-19 and 2021-09-16 events were the largest and most intense events. The antecedent conditions were dry for the event on 2021-09-16, but wet for the event on 2021-09-19. Furthermore, the event on 2021-09-16 had a shorter duration and higher intensity leading to a very large increase in streamflow and expansion of the drainage network length ($\Delta Q = 1140$ and 1634 [l min$^{-1}$] and $\Delta fFDNL = 0.78$ and 0.36 for Lan and Cha, respectively; Table 1).

The isotope-based hydrograph separation results for the events 2021-09-10, 2021-09-29, and 2021-10-21 indicated maximum event water fractions ($f_e$, max) between 0.20 and 0.48 for Lan and between 0.13 and 0.26 for Cha (no data for 2021-10-21; Table 1). The maximum event-water fractions were correlated with the changes in flowing network length (for Lan: r=0.99, p=0.019, n=3). Notably a one-fold change in $f_{FDNL}$ was accompanied by a ~1.5-fold change in the maximum event-water fractions.


Table 1: Overview of the seven events that were sampled. Events are ranked by the total event precipitation P [mm]. $Q_{48}$ is the average streamflow in the two days before the event [l min$^{-1}$]; $P_{48}$ is the total precipitation in the two days before the event start [mm]; $GWL_{48}$ is the groundwater levels measured at location E2 (in cm below the surface), Q/P is the unitless runoff coefficient (event total streamflow per unit area divided by the total precipitation); $\Delta Q$ is the increase in streamflow during the rainfall-runoff event, i.e., the difference between streamflow at the time of the event start and the peak [l min$^{-1}$]; $T_{peak}$ is the time between rainfall start and peak discharge (hours); Duration is the time (hours) between event start and end (see section 2.5.1); $f_e$, max the maximum event water fraction (± standard error) [-]; and $\Delta f_{FDNL}$ is the change in the $f_{FDNL}$. The event on the 21$^{st}$ of October was sampled only in Lan.

| Indices | Variable | Units | Site | Sep 10 | Oct 21 | Sept 29 | Oct 12 | Aug 28 | Sep 19 | Sep 16 |
|---|---|---|---|---|---|---|---|---|---|---|
| **Rainfall event** | P | mm | Both | 11 | 15 | 16 | 23 | 28 | 31 | 34 |
| | Q/P | - | Lan | 0.05 | 0.20 | 0.20 | 0.39 | 0.42 | 0.41 | 0.31 |
| | | | Cha | 0.50 | 0.35 | 0.50 | 0.34 | 0.7 | 0.45 | 0.38 |
| | $\Delta Q$ | l·min$^{-1}$ | Lan | 16 | 370 | 116 | 529 | 707 | 686 | 1140 |
| | | | Cha | 408 | 891 | 219 | 57 | 648 | 833 | 1634 |
| | $T_{peak}$ | h | Lan | 1 | 3 | 24 | 12 | 23 | 7 | 8 |
| | Duration | h | Both | 75 | 55 | 115 | 126 | 123 | 99 | 66 |
| | $f_{e, max}$ | - | Lan | 0.20 ± 0.03 | 0.48 ± 0.05 | 0.34 ± 0.03 | - | - | - | - |
| | | | Cha | 0.26 ± 0.05 | - | 0.13 ± 0.03 | - | - | - | - |
| | $\Delta f_{FDNL}$ | - | Lan | 0.32 | 0.70 | 0.52 | 0.61 | 0.58 | 0.51 | 0.78 |
| | | | Cha | 0.33 | 0.39 | 0.26 | 0.15 | 0.28 | 0.25 | 0.36 |
| **Antecedent wetness** | $P_{48}$ | mm | Both | 2 | 0 | 0 | 0 | 3 | 0 | 4 |
| | $Q_{48}$ | l·min$^{-1}$ | Lan | 2 | 3 | 2 | 5 | 8 | 30 | 3 |
| | $GWL_{48}$ | cm | Both | -157 | -142 | -149 | -135 | -142 | -121 | -162 |
| | Conditions | - | Both | dry | dry | dry | wet | wet | very wet | dry |

### 3.2.2. Description of the flowing drainage network and stream chemistry changes during the 2021-09-29 event

In this section, we describe the hydrological and hydrochemical responses in the two catchments before and during the 16 mm rainfall event on 2021-09-29 (Fig. 4). We selected this event because it was the largest of the three events for which hydrograph separation was possible for both catchments. Moreover, most of the precipitation in the Erlenbach has a low intensity, i.e., 10-min precipitation < 3 [mm h$^{-1}$] (van Meerveld et al., 2018). Therefore, we used the event on 2021-09-29 as an emblematic example of how the flowing drainage network and solute concentrations change during small events with dry antecedent conditions. Figure 4 provides a detailed insight into the spatiotemporal variations in the hydrometric response (i.e., changes in streamflow, groundwater levels, and $f_{FDNL}$) and streamwater chemistry, including the event water contributions. Similar figures for the other events can be found in the supplementary materials (Fig. S3-S8).

The flowing drainage network in Lan was fragmented and short ($f_{FDNL}$=0.09). The streams were flowing close to the catchment outlet, as well as locally in the middle of the catchment (but note that the network was disconnected, i.e., streamflow infiltrated into the subsurface further downstream; $t_1$ in Fig. 4a, Fig. 6a). In Cha, groundwater emerged from perennial springs in the

central part of the catchment at the bottom of the steep hillslope (compare Fig. 1 and Fig. 4, Fig. 6c) and the continuous flow from this spring caused the specific discharge in Cha and $f_{FDNL}$ (0.6 vs 0.09) to be higher than in Lan (Fig. 4e).


Before rainfall started (i.e., until time step $t_1$ in Fig. 4), streamwater chemistry at the outlets of Lan and Cha was typical of baseflow conditions during the study period (Fig. 2), i.e., the concentrations of weathering-derived solutes (e.g., calcium, silica, sulfate) were high (Fig. 6g) and concentrations of nitrate, chloride, and iron were low (Fig. 3 and Fig. 4b, Fig. 6e,h,f). This is consistent with our expectation that groundwater is the main source of streamflow during baseflow conditions (Fig. 6a,c, dark

blue areas). The concentrations of chloride and iron were higher in Lan than Cha, likely due to local groundwater sources that are rich in these solutes (e.g., LGW3, Fig. 2, see section 3.1.1). The end member mixing analyses based on silica and $\delta^2H$ suggest that soil water did not contribute to baseflow.

After the onset of rainfall, $f_{FDNL}$ in Lan increased within approximately five hours from 0.09 to 0.42 ($t_1$-$t_2$ in Fig. 4) because

more uphill channels started to flow (Fig. 6b). The drainage network did not expand to its full extent during this event, likely because the groundwater levels did not reach the more permeable rooting zone at around 0-20 cm depth, e.g., the water level in LGW1 rose from 42 to 23 cm below the soil surface. The event water fraction ($f_e$) increased within six hours to 0.34, i.e., for slightly longer than $f_{FDNL}$. Assuming an on average 0.2 m wide stream, the rainfall falling directly on the flowing stream network was larger than the observed event water flux (due to the very small event water fraction at this time) and even the

observed increase in streamflow when the intensity of rainfall was high (1:30-2:30) and shortly after, i.e., for approximately first two hours of the event (Fig. S13). This suggests that at the start of the event, rain falling on disconnected stream segments did not immediately make it to the outlet and that some water will have infiltrated into the channel. However, afterward, the calculated event water flux was always larger than the discharge that could have been produced by the rain falling on the channel. Over the entire event, the total event water flux was more than six times the amount of rainfall that fell on the channel.

Interestingly, the $f_{FDNL}$, $f_e$, and the groundwater level remained high for more than ten hours after rainfall stopped, whereas discharge decreased from 34 [l min$^{-1}$] to 14 [l min$^{-1}$] ($t_3$ to $t_4$, Fig. 4b). This suggests that other areas must have contributed event water to streamflow as well (Fig. 6b,c), e.g., saturated overland flow or macropore flow through the topsoil.

In Cha, the flowing drainage network expanded rapidly from 0.60 to 0.84 in less than one hour after rainfall started ($t_1$ to $t_3$ in

Fig. 4e). Contrary to Lan, $f_{FDNL}$ in Cha returned to the pre-event conditions within 1.5 hours after the rainfall stopped. The contribution of event water in Cha increased as quickly as in Lan, but peak $f_e$ was much smaller ($< 15\%$ at $t_3$), likely because the geomorphic drainage network in Cha was generally much shorter than in Lan (15.2 [km$^{-1}$] vs 40 [km$^{-1}$]). The groundwater levels in GBGW1 were still low at the time that peak flow occurred (at about 53 cm below the surface) but kept rising slowly afterward (to about 50 cm below the surface). Discharge and $f_{FDNL}$ returned quickly to pre-event conditions ($t_3$ to $t_4$ in Fig. 4e).


The concentration of most weathering derived solutes were similar to those expected from the simple mixing of event water and pre-event water for both Lan and Cha (Fig. 4b,e). Concentrations of iron and chloride increased, suggesting a connection to groundwater from other stores and interflow in topsoil (Fig. 6f,h). During the time of the rapid increase in streamflow and $f_{FDNL}$ in Lan, chloride and nitrate concentrations increased substantially until shortly after the peak flow and before peak $f_{FDNL}$.

Such a response is consistent with the mobilization from dry stream channels or the material that was stored in them. Unlike the flushing observed in Lan, nitrate concentrations in Cha streamwater decreased slightly during the event (Fig. 6e). This was likely due to Cha's shorter channel network with only a small portion of it falling dry, so that there was less nitrate mobilization from dry stream channels or transport of material that was stored in them, compared to Lan. The observed differences in the dynamics of solute concentrations and flowing drainage network expansion between Lan and Cha underscore how changes in

the flowing drainage density and the onset of flow in previously dry channels may influence catchment solute responses.

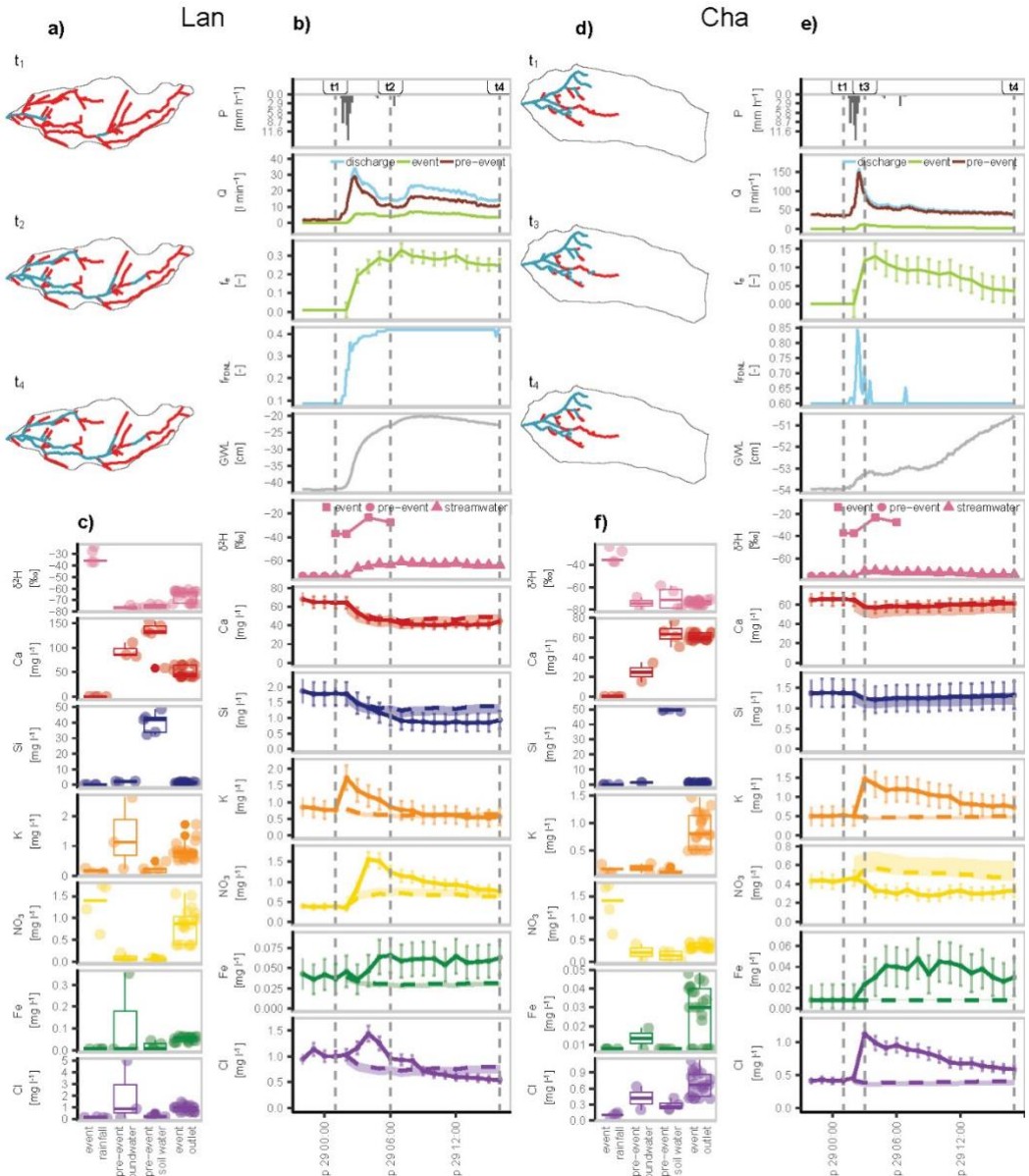

Figure 4: Maps of the flowing (blue) and not flowing (red) reaches of the channel network during three selected times during the 2021-09-29 event (a, d); time-series of hydrologic variables, $\delta^2H$ and solute concentrations, and the results of the hydrograph separation (b, e); box plots of the concentrations of the samples collected at the catchment outlet during the event, the weighted mean of the rainfall sampled during the event, pre-event groundwater samples, and pre-event soil water (lysimeter) samples (c, f) for the Lan (left) and Cha (right) catchments. In panels a, b, d and e, $t_1$ represents the start of the event, $t_2$, and $t_3$ mark the time of the maximum extension of the flowing drainage network (i.e. the largest $f_{FDNL}$), and $t_4$ marks the time when the last sample was collected during the event. The dashed line in the time series of b and e represents the expected concentration based on the hydrograph separation results, with the shaded areas indicating the upper and lower bound of the expected concentrations based on the standard error of $f_e$. GWL in panels b and e stands for groundwater level at locations LGW1 (Lan) and CBGW2 (Cha). Boxes in panels c and f show the 1st, 2nd, and 3rd quartiles and whiskers extend to the minimum and maximum values. The data for the other six events are presented in the supplementary material (Figures S3-S8).

### 3.3. Inferences about runoff generation mechanisms in the Lan catchment

#### 3.3.1. Rainfall on flowing channels and flow from saturated areas

The event water fraction increased with the fraction of the flowing drainage network (Fig. 5a) and varied from 0 to 0.9 for the three events for which hydrograph separation was possible for the Lan catchment. Moreover, both $f_{FDNL}$ and $f_e$ were higher when streamflow increased more during the events ($\Delta Q$; Table 1). During the first two hours of the events, event water fractions increased slowly (i.e., $f_e < 0.2$; Fig. 5a), but they increased more rapidly once $f_{FDNL}$ was higher than 0.4 (Fig. 5a), which marks the transition from a fragmented (Fig. 6a) to a connected (Fig. 6b) drainage network (Bujak-Ozga et al., 2023). This suggests that the connection of the flowing drainage network caused an increase in the contribution of event water at the catchment outlet. However, direct precipitation on the flowing stream channel was much less than the calculated event water flux (Fig. S13-S16). Furthermore, event water fractions remained high even after rainfall ceased and discharge decreased (Fig.4, Fig. S3-S8). This indicates that event water must have originated from areas outside the channel network, such as saturated overland flow and shallow (root zone) interflow. Saturated overland flow has been observed across many locations in the neighboring Studibach catchment, particularly in the wet meadow areas (van Meerveld et al., 2019). Infiltration excess overland flow likely plays a minor role as the soil surface hydraulic conductivity typically exceeds the rainfall intensities (van Meerveld et al., 2019; Wadman 2023).

The increased concentrations of iron during rainfall events are likely caused by the transport of more soluble and mobile ferrous iron minerals from the reduced conditions of saturated areas (typical for gleysols; Hewitt et al., 2021) that become connected during rainfall events and start contributing to streamflow at the outlet. Iron concentrations increased sharply from circa 0.02 to 0.9 [mg l$^{-1}$] when $f_{FDNL}$ increased to 0.5 and the network became connected (Fig. 5s). This increase was consistent for all events (Fig. 6f). When $f_{FDNL}$ in Lan exceeded 0.5, surface flow occurred in multiple shallow channels and artificial ditches, highlighting the saturation of the system as well (Bujak-Ozga et al., 2023). Moreover, iron concentrations were the most variable at low discharge, and we observed a slight mobilization when the network became connected. When shallow channels and artificial ditches were flowing, iron concentrations were high, varied less, and showed chemostatic behavior (Fig. S12q). Previous research suggested that a decrease in the coefficient of variation in solute concentrations with increasing discharge can be caused by a higher connectivity due to the wetter conditions and spatial variations in solute concentrations within the catchment (Knapp et al., 2022). Thus, the high spatial heterogeneity in iron concentrations could contribute to the observed CQ relationship for iron in Lan. Indeed, we observed iron concentrations of around 0.7 [mg l$^{-1}$] in one of the groundwater wells in Lan (LGW1; Fig. 3), suggesting that locally higher iron concentrations are present in Lan, depending on the redox conditions. This is supported by previous studies in the Lan area (Hagedorn et al., 2001) and the neighboring Studibach catchment (Kiewiet et al., 2019) that found high iron concentrations in soil and groundwater in waterlogged depressions and

riparian-like areas, and lower concentrations in aerobic mounds and hillslope groundwater. Similar results have also been found in other catchments. For example, research in over eighty German catchments showed that the mean iron concentrations in streamwater were high in flatter catchments where shallow groundwater tables create conditions favorable for reductive processes (Tittel et al., 2022).

### 3.3.2. Groundwater flow

Pre-event water was the dominant source of streamwater at the Lan catchment outlet (Fig. 5a). The high pre-event water fractions (Fig. 5a) during the first two hours of the events suggest that the increase in discharge was mainly related to an increase in pre-event water (e.g., Fig. 4b, Fig. S3b-S8b). The considerable increase in pre-event water during the event suggests that the infiltrating rainfall caused additional groundwater to be released (cf. the low event water contributions as a fraction of precipitation observed by von Freyberg et al., 2018).

The increase in pre-event water at the catchment outlet is likely controlled by the rise of the groundwater levels into more permeable topsoil layers and the expansion of the groundwater contributing source area (cf. Rinderer et al, 2019). Previous research has suggested that vertical concentration profiles of weathering-derived solutes can cause dilution patterns in streamwater as well (Botter et al., 2020; Knapp et al., 2022; Seibert et al., 2009). In Lan, calcium concentrations in the shallow soils increase with depth below the surface (Hagedorn et al., 2001). This was also observed in the neighboring Studibach catchment (Kiewiet et al., 2020; Bruppacher, 2022). Vertical stratification of weathering-derived solute concentrations in soil and groundwater could potentially lead to a decrease in concentrations at the outlet as the groundwater levels increase during rainfall events and water with lower concentrations in shallower soil layers is transported and becomes hydrologically connected to the stream. However, this process likely plays a minor role in Lan because the observed concentrations for weathering-derived solutes (calcium, sulfate, sodium, magnesium, silica) correspond well with those expected from a simple mixing with event water (e.g., Fig. 4). This suggests that the primary driver for these changes in solute concentrations is the dilution of the groundwater with event water, rather than contributions from different parts of the vertical soil profile. Furthermore, the time of the groundwater response and dilution patterns did not coincide for many events (e.g., Fig. S7). Finally, the large spatial variation in groundwater chemistry may mask the effect of any vertical variation (Fig. 3).

### 3.3.3. Soil water and interflow

The high concentrations of silica and magnesium in soil water (lysimeter samples) were used to infer the importance of interflow in the subsoil (i.e., flow at ca. 30 cm from the soil surface). The dilution patterns for these solutes were similar to those of the weathering-derived solutes (e.g., Fig 4. and Fig. 5i,k). This suggests a very small contribution from soil water at the Lan catchment outlet, which is further supported by the EMMA results that indicated that $f_{sw}$ was negligible. Similar results

were found by Kiewiet et al. (2020) for the Studibach catchment, where pre-event groundwater was the dominant source of streamflow and soil water contributions were minimal for three out of the four analyzed events.

We observed more pronounced clockwise hysteresis for potassium and chloride (Fig. 5m,o; Fig. S11a; Fig. S10g) than for the other solutes. The increase in potassium and chloride concentrations at the catchment outlet (i.e., above the expected

concentration according to the simple mixing of baseflow and rainfall; Fig. 4b and Fig. S8b) indicates that additional sources that are rich in those nutrients contributed to the streamflow. Relatively high potassium and chloride concentrations were assumed to be indicative of contributions from soil water in the Studibach in previous studies (Kiewiet et al, 2020). A study of vertical concentration profiles in soils in the Studibach catchment (Bruppacher, 2022) highlighted that potassium concentrations were the highest in the shallow soil (12.5 cm below the surface). Thus, the observed hysteresis in potassium

concentrations could be caused by a contribution of water from the shallow topsoil (i.e., above the installed lysimeters). However, they could also have been caused by the rising groundwater levels and expansion of the groundwater contributing area because groundwater in Lan was highly variable in chloride and potassium concentrations (Fig. 3), which is typical for the Alptal catchments (Kiewiet et al. (2019); Knapp et al., 2020). Due to this high spatial variability (Fig. 3), additional measurements are needed to determine if the changes in chloride and potassium concentrations are caused by the contribution

of shallow soil water and/or different groundwater source areas.

Knapp et al. (2020) also reported similarity in the potassium and chloride dynamics and predominantly mobilization of chloride during events for the Erlenbach catchment. They also reported a larger slope of the CQ relationship for large events during dry conditions, which agrees with the differences in the CQ relation observed for the 2021-09-16 and 2021-09-19 events (Fig. 5o).

However, there were also differences in the response timing (e.g., Fig. 4b) and/or concentrations (e.g., Fig. S8b). Chloride concentrations were less variable at lower discharge than at higher discharge (Fig. S12q). Potassium concentrations are affected by biological processes, which could contribute to its general high variability (Fig.S12o), especially for the events with dry antecedent conditions (Fig. S10g; 2021-09-10).

**3.3.4. Flush of nitrate from previously dry channels**

Nitrate concentrations in streamwater in Lan increased more than expected from the mixing of event and pre-event water (Fig. 4b, Fig. S3b-S8b), suggesting a mobilization of nitrate (Fig. 5q). The increase in nitrate concentrations occurred during all rainfall events but the magnitude of the change varied between events. For the smallest event during dry conditions on 2021-09-10 (Table 1), streamwater nitrate concentrations increased by only 0.3-0.8 [mg l$^{-1}$]. During this event, many channels

remained dry, and f$_{FDNL}$ (max f$_{FDNL}$: 0.4) and streamflow remained low ($\Delta Q$: 16 [l s$^{-1}$]). There was especially a lack of response in the reaches with a low flow persistency. The largest event (2021-09-16), which occurred during moderately dry antecedent conditions, caused the expansion of the full network. For this event, nitrate concentrations were much higher and increased up

to 3 [mg l$^{-1}$]. For the event on 2021-09-19, only three days later, with wet antecedent conditions, an almost full expansion of the FDNL ($f_{FDNL}$: 0.9), and the second highest peak discharge, nitrate concentrations remained below 1 [mg l$^{-1}$]. Such inter- and intra-event variability in nitrate concentrations is not surprising. The nitrate pool was likely already mobilized during the preceding event and there had not been enough time for a new nitrate pool to accumulate. The two other events that were also characterized by relatively high nitrate concentrations (2021-10-12 and 2021-10-21) occurred in October when the vegetation started shedding leaves and more detritus accumulated on the surface and in the dry channels. Thus, we attribute the nitrate flush to the transport of material from the previously dry channels. Nitrate "flushes" following the reconnection of flow have been observed in other studies in intermittent streams (e.g., von Schiller et al., 2011; Merbt et al., 2016). Mineralization of organic matter from accumulated detritus and nitrification in the streambeds during the dry phase are a main source of inorganic nitrogen in streams (Baldwin et al., 2005, Woodward et al., 2015, Merbt et al., 2016; von Schiller et al., 2017; Arce et al., 2018). Moreover, Merbt et al. (2016) showed that ammonia oxidation in dry streambeds can contribute to roughly 50% of nitrate flush in intermittent streams. The drainage network in Lan has a large number of reaches that fall dry between rainfall events (i.e., that have a low flow persistency), allowing for nitrate accumulation in the dry streambeds (Fig. 6a).

Accumulation of nitrate can also occur in soils, but both groundwater and soil water in Lan were characterized by low nitrate concentrations (Fig. 2, Fig. 4c). Thus, groundwater and soil water are unlikely to contribute to higher nitrate concentrations at the catchment outlet. Kiewiet et al. (2019) showed that in the neighboring Studibach catchment, median nitrate concentrations in groundwater were also low (<0.1 [mg l$^{-1}$]. A lack of nitrate in deeper soil horizons is typical for gleysols (Hewitt et al., 2021) due to the high/perched groundwater tables and anaerobic conditions. Hagedorn et al. (2001) found that in areas close to Lan, nitrate was present only in the topsoil (0-10 cm depth), and even there only in relatively low concentrations (median concentrations < ~0.5 [mg l$^{-1}$]). In the deeper soil layers, nitrate concentrations were below the detection limit.

Dry deposition can be a source of nitrate in forested catchments and could be the reason that a few rainfall samples taken at the beginning of the events with higher nitrate concentrations (Fig. S7, Fig. S8). Concentrations in throughfall are higher (median: ca. 2.4 [mg l$^{-1}$]) than for precipitation (Hagedorn et al., 2001). Considering the high percentage of forest in Lan, throughfall could be a potential source of nitrate for Lan. However, nitrate concentrations at the Lan catchment outlet were highest when segments with low flow persistency were flowing (Fig. S12s), and nitrate concentrations remained high even after rainfall ended (Fig. 4b, $t_3$-$t_4$). Furthermore, nitrate concentrations increased less for events for which the stream network was already fully extended or had recently had flowing water. Therefore, we think that a nitrate "flush" due to mobilization from the previously dry streambeds is a more important reason for the observed nitrate dynamics at the catchment outlet than the contribution from throughfall.

### 3.3.5. Summary of dominant flow processes in Lan

In the flatter Lan catchment, flow is absent in a large fraction of the drainage network during dry conditions, especially in shallow channels and artificial ditches, and the network is disconnected (Fig. 6a, red lines). The FDN remains short and fragmented (Fig. 6a, dark blue lines) until shortly after rainfall begins, with flow consisting of pre-event water fed by groundwater (Fig. 6a, dark blue area). As rainfall continues (Fig. 6b), the FDN expands rapidly (Fig. 6a, light blue lines), connecting previously dry channels and saturated areas and new areas become saturated (Fig. 6b, light blue areas). The transition from a fragmented to a connected network affects the transport of rainfall and overland flow to the catchment outlet. At the time of the connection of the network, there is a considerable increase in the contribution of event water at the catchment outlet, leading to rapid dilution of weathering derived solutes (see step change in Fig. 5 and also Fig. 6g). The re-emergence of flow in channels and the expansion of the drainage network also leads to a nitrate "flush" from previously dry channels (Fig. 6e) but this flush depends on the antecedent wetness conditions and event size. Later during the events, other source areas become connected to the stream and start to contribute to streamflow. This is especially the case during wetter conditions and larger events (Fig. 6b, light blue areas). Increased connectivity between these source areas due to the rising water table and connectivity between these source areas and the channel due to the expanded network during wetter conditions leads to the mobilization of iron from saturated areas (Fig. 6b, and 6f). Saturated overland flow or lateral flow of event water through the macropores in the topsoil are an important source of streamflow and are the likely source for the increases in potassium and chloride concentrations (Fig. 6h) during events. These near surface flow pathways must also provide a considerable amount of event water to the stream as the total amount of event water can not be explained by rainfall falling on the flowing stream network. The role of interflow deeper in the soil (i.e., the subsoil) during rainfall events is likely minor, as most of the interflow occurs in the more permeable topsoil.

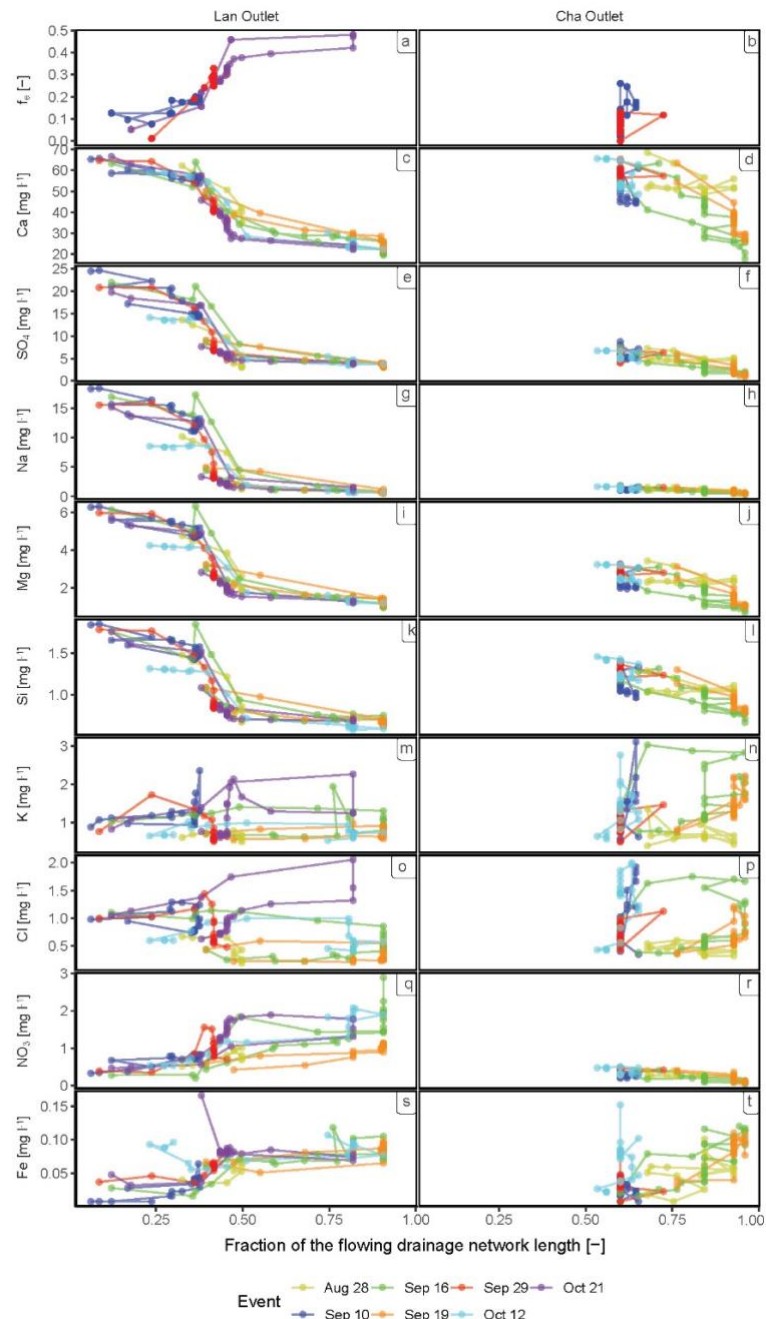

**Figure 5: Event-water fractions ($f_e$) and solute concentrations plotted as a function of the fraction of the drainage network that was flowing ($f_{FDNL}$) at the time that the sample was collected for the Lan catchment (left-hand panel) and the Cha catchment (right-hand panel). The different colors represent the different events. One chloride measurement (12-10-2021: 4.68 [mg l$^{-1}$]) and two potassium measurements (2021-09-11: 5.66 [mg l$^{-1}$], and 3.99 [mg l$^{-1}$]) for samples from Lan plot outside the axis ranges and are not shown to increase the visibility of the other data points. For the graphs showing the event-water fractions ($f_e$) and solute concentrations as a function of the discharge see Fig. S11, for the graphs showing the concentrations as a function of the event-water fractions, see Fig. S12.**

### 3.4. Inferences about runoff generation mechanisms in the Cha catchment

#### 3.4.1. Rainfall on flowing channels and flow from saturated areas

The flowing drainage network in Cha was more constrained than in Lan: $f_{FDNL}$ changed only between 0.5 and 0.9 (Fig. 5). There was also no apparent rapid transition between a fragmented and connected network. The event water fractions could be calculated for two relatively small events but were different from those in Lan (Fig. S11 and Fig. S10). For these small rainfall events, event water played only a minor role in streamflow generation. Unlike Lan, event water fractions were not well related to the fraction of the flowing drainage network (Fig. 5b). The correlation of the event water fractions with the discharge was also poorer than for Lan (Fig. S12b).

The CQ relation for iron, for example, showed chemodynamic behavior for the small events, and mobilization for the bigger events with the largest increases in streamflow (Fig. 5t, S12s, Table1). Increasing iron concentrations and decreasing nitrate concentrations in streamflow could suggest an increased input of water from the saturated area. The drainage network in Cha is located in the western, flatter part of the catchment. This downstream part of the catchment is characterized by a wet area with shallow water tables, which is typical for gleysols (Hewitt et al., 2021; Rinderer et al., 2014). The high groundwater levels and anoxic conditions are favorable for nitrate removal and can explain the higher iron and lower nitrate concentrations in soil and groundwater in this part of the catchment (CBGW1 and CBGW1LYS; Fig. 2) than the steeper, upstream, eastern part of the catchment (CBGW2 and CBGW2LYS; Fig. 2), where there is probably more infiltration into the better drained drier soils and nitrate removal due to the vegetation uptake. Iron concentrations started to increase ca. two hours earlier in Cha than in Lan, but the streamflow response was generally faster for Cha than Lan as well. The faster response in Cha can simply be related to the spatial distribution of the source area within catchments (Knapp et al., 2022), or the steeper slope of the catchment.

#### 3.4.2. Groundwater flow

The pre-event water fractions were higher in Cha than Lan. Sulfate, magnesium, and sodium concentrations in baseflow were relatively low, suggesting that the groundwater that feeds the perennial springs is not very old. Calcium dissolves quickly, thus, concentrations in baseflow were comparable to Lan. Similar to Lan, the observed variations in the concentrations of weathering-derived solutes (Ca, Na, $SO_4$, S, Mg, Si) matched the expected concentrations based on simple mixing of pre-event water and event water. However, for larger events, there was hysteresis for calcium, magnesium, and silica (Fig. 5d,j,l). Moreover, the variability in the concentrations of weathering-derived solutes in Cha was higher at higher discharges (Fig. S12). In Lan, the response was dominated by dilution, but in Cha, the concentrations appear to be also effected by the

connection of the upper hillslope to the stream, which likely resulted in more variability in the measured concentrations at the outlet (as also indicated by the variability of the concentrations of groundwater, lysimeter, and baseflow samples taken across the catchment (Fig. 3). We, furthermore, observed more inter-event variability in calcium concentrations than for sulfate and sodium in Cha. For the biggest event (2021-09-16), counter-clockwise hysteresis could be observed (Fig. 5d). Since shallow groundwater in Cha is depleted in calcium, such patterns might be indicative of shallower flowpaths. This is further supported by a small hysteresis in silica and magnesium concentrations during events (Fig. 5j,l) as we found higher concentrations of these solutes in soil water than groundwater.

### 3.4.3. Soil water and interflow

Similarly to Lan, concentrations of silica and magnesium in soil water in Cha showed a dilution pattern (e.g., Fig. 5j,l). The generally small contribution from soil water at the Cha catchment outlet was further supported by the results of the EMMA analysis for the two smaller events (calculated contributions from soil water: 0%). Also, similar to Lan, chloride, and potassium showed a mobilization pattern (Fig. S12p,r) and increased more than could be expected based on simple mixing of rainfall and baseflow (Fig. 4e and Fig. S8e). Moreover, chloride and potassium concentrations in Cha showed very similar clockwise hysteresis during rainfall events, which was more pronounced than for Lan (Fig. 5n,p; Fig. S11b; Fig. S10h). This could suggest that the same processes, i.e., interflow in the topsoil or a connection to groundwater sources with high chloride and potassium concentrations are responsible for the chloride and potassium dynamics in both catchments. After $f_{FDNL}$ and $f_e$ reached their maximum, potassium, and chloride generally decreased and returned to their expected values or were slightly lower. These decreases tended to be faster for Lan than in Cha. This quick response suggests that a contribution from shallow flow pathways is a more likely source of chloride and potassium than connectivity to a different groundwater source.

### 3.4.4. No nitrate flush

The nitrate response in Cha was very different from that observed in Lan. There was no clear nitrate flush in Cha and nitrate concentrations were lower than expected from the simple mixing of event and pre-event water (Fig. 4e). Nitrate concentrations in Cha did not exceed 1 [mg l$^{-1}$], while in Lan, concentrations during events went up to ca. 3 [mg l$^{-1}$] (Fig. 5-7). Nitrate concentrations in soil and groundwater were low for both Cha and Lan (Fig. 2-3) and if anything one could expect the concentrations of nitrate to be slightly higher in Cha than in Lan because of the additional inputs from grazing cattle and less nutrient uptake by vegetation. The difference in nitrate dynamics between Lan and Cha are therefore likely related to the difference in the length of the channels that are dry for considerable periods (i.e., segments with low flow persistency). In Cha, there are fewer channels in which nitrate could accumulate and subsequently be flushed out. Furthermore, there are fewer channels where direct precipitation, which is relatively high in nitrate (Fig. 3) can directly contribute to runoff. Instead, as discussed above, there is dilution of nitrate due to the contribution of runoff from saturated areas that are low in nitrate.

### 3.4.5. Summary of dominant flow processes in Cha

In the steeper Cha catchment with the short channel network and perennial springs, groundwater flow is the dominant source
of streamflow at the catchment outlet before and during events (Fig. 6c). Contrary to Lan, the springs, located at the break in
slope maintain a higher baseflow and a more stable and connected flowing drainage network (Fig. 6d). The expansion of the
flowing drainage network is very limited and only occurs in the lower part of the catchment (Fig. 6d, light blue lines), leading
to more gradual changes in the event water contributions and solute concentrations (Fig. 6g).

Because only very few channels become activated during rainfall events (Fig. 6d, light blue lines), there is no nitrate "flush"
during events. Instead, nitrate is diluted due to the increasing contribution of runoff from saturated areas (Fig. 6e). During
small events with dry antecedent conditions, the smaller contribution of event water than in Lan leads to smaller changes in
solute concentrations (Fig. 4). Similarly to Lan, the increased connectivity during wetter conditions leads to the connection of
saturated areas to the channel network and the transport of iron from these saturated areas (Fig. 6d,f). Similar to Lan, saturated
overland flow or lateral flow through the macropores in the topsoil are the likely causes for the increases in the event water
noncontributions during larger events and the more extensive dilution of weathering derived solutes during large events. These
flow pathways are also responsible for the increases in the concentrations of potassium and chloride (Fig. 6h). The shorter
stream network concentrated close to the outlet likely results in the quicker connectivity between saturated areas and the
channel network and the faster response time for solutes like potassium and chloride compared to Lan.


### 3.5. Flowing drainage network and solute export

The differences in topography, total drainage network length, flow persistency, and connectivity of the flowing parts of the
channel network for the Lan and Cha catchments highlight the role of stream network dynamics (and thus geomorphology) on
solute transport. Expansion of the flowing stream network increases connectivity between different parts of the drainage
network and can cause the transport of material stored in dry channels. For the flatter Lan catchment, the transport of nitrate
from previously dry channels was important but this was not the case for the Cha catchment for which the flowing channel
network was more stable. The connection of the previously fragmented drainage network in Lan led to a sudden increase in
the contribution of event water and dilution of weathering derived solutes. The changes in the event water fraction and
concentrations of weathering derived solutes were more gradual for the Cha catchment where the changes in the flowing stream
network were smaller.

The expansion of the flowing stream network also increased the connectivity between the channels and hillslopes. The
increased connectivity between the channels and the saturated areas facilitated the transport of water from overland flow in

saturated areas and flow through macropores in the topsoil to the stream, increasing concentrations of iron, chloride and

potassium. Although these flow pathways were important for both catchments, the timing of the contributions of these near surface flow pathways was different for Lan and Cha. This is likely related to the different catchment characteristics, such as the location of the contributing areas relative to the channel network and the slope. Of course, the differences in the length of the channel network and overall slope are not the only factors that influence the relative contributions of different flow pathways or hillslope-stream connectivity for the Lan and Cha catchments, as differences in vegetation cover, soil depth, etc.,

can also influence these interactions. Therefore, the changes in stream network dynamics and solute concentrations need to be studied for a range of events in other catchments as well.

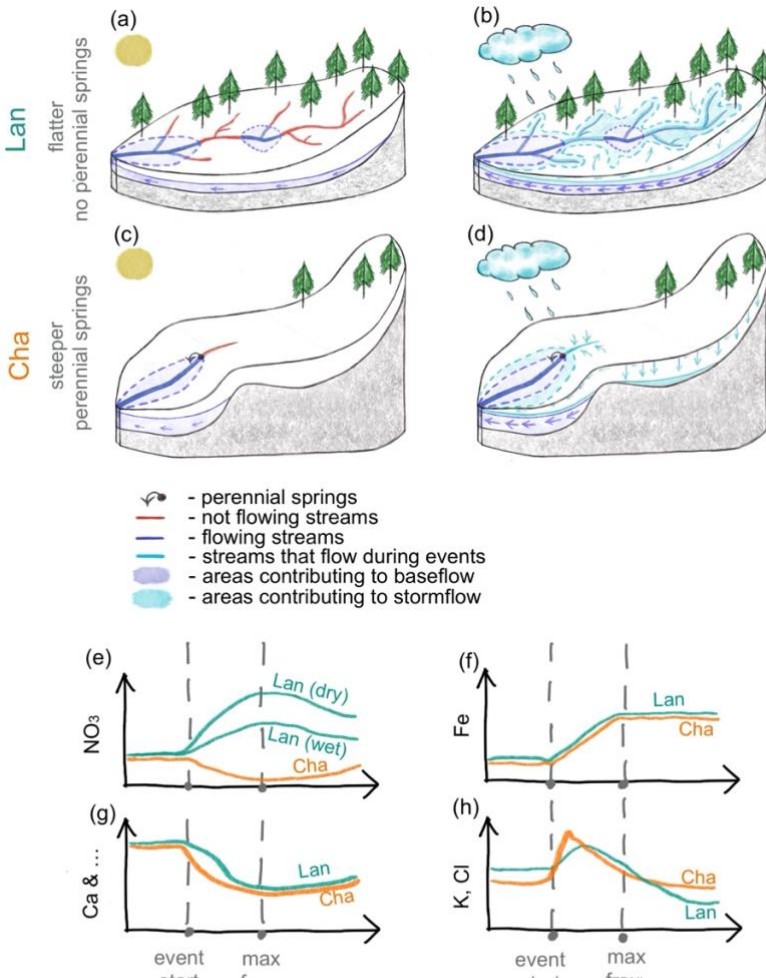

Figure 6: Conceptual diagram showing the flowing drainage network in Lan and Cha and the main processes leading to streamflow during baseflow conditions (a, c) and rainfall events (b, d). The related temporal variations in solute concentrations at the catchment outlet are shown for nitrate (e), iron (f), calcium (as an example of weathering-derived solutes) (g), and potassium and chloride (h). The blue and orange lines in e-h represent the solute dynamics for Lan and Cha, respectively. Solid lines on the surface in a-d represent channels without flow (red), channels that were already flowing before the start of the event (dark blue), and channels that started to flow during event (light blue). The dashed lines represent source areas that contributed to streamflow before event (dark blue; groundwater), and during events (light blue; groundwater, saturated overland flow, and interflow through the topsoil). The two green lines for the nitrate dynamics in e represent the different magnitude of response in Lan for events with dry ("flush") and wet antecedent conditions. In Lan (with flatter topography and longer channel network), the flowing drainage network expands rapidly during rainfall. This leads to a rapid increase in connectivity of the flowing drainage network, which shapes the solutes responses, especially for weathering derived solutes. Moreover, re-emergence of flow in previously dry channels leads to an increase in nitrate concentrations at the outlet ("flush") that depends on the antecedent conditions. In contrast, the more stable flowing drainage network in the steeper Cha results in more gradual changes in solute concentrations and no nitrate flush. The connection of saturated areas to the channel network facilitates the transport of iron from these saturated areas.

## 4. Conclusions

In this study, we investigated how solute concentrations, event water fractions, and the flowing drainage network change during rainfall events in two small headwater catchments of similar size, with similar soils and bedrock. In both catchments, groundwater was the main source of streamflow and the dynamics of the concentrations of the weathering-derived solutes could be explained by the mixing of event water and pre-event water (Fig. 4). At the beginning of rainfall events, event water contributions could be explained by rain falling onto the channels. However, as the events progress, this is insufficient to account for the observed event water flux. Thus, additional contributions from overland flow from saturated areas (Fig. 6) or quick interflow through the topsoil are also important, particularly under wetter conditions when saturated areas become connected to the stream. The dynamics of the potassium and chloride concentrations suggest a contribution from interflow from the topsoil as well. The contribution from the deeper (~30 cm) subsoil were minor.

However, there were also distinct differences in the hydrological and hydrochemical responses of the two catchments. In the flatter Lan catchment, the flowing drainage network expanded rapidly and up to 10-fold during rainfall events. Flow in previously dry channels caused the mobilization of nitrate. In the steeper Cha catchment with a shorter and more stable flowing drainage network, nitrate was diluted during events due to the contribution of runoff from saturated areas (Fig. 6).

Overall, this study contributes to the growing body of scientific literature on intermittent streams and the associated dynamics in stream chemistry (e.g., Hale and Godsey, 2019; Zimmer and McGlynn, 2018; von Schiller et al., 2017; Merbt et al., 2016). It highlights the high variability in the flowing stream network and stream chemical responses for neighboring catchments and how information on the flowing stream network can help to understand stream chemistry dynamics and interpret runoff processes. It also underscores the high variability in the hydrochemical response of neighboring catchments and the need to consider geomorphic characteristics when interpreting stream solute responses for different catchments, as well as the larger catchments to which these small headwater catchments contribute.

## Data availability

Hydrometric (Bujak-Ozga et al., 2023b) and hydrochemical (Bujak-Ozga et al., 2024) data from this study are available on EnviDat.ch, where most of the data from the Alptal catchments are stored.

## Author contribution

IBO, IvM and JvF conceptualized the study; IBO collected, cured and visualized the data; IBO analyzed and interpreted the data with support from all co-authors; IBO wrote the manuscript draft; IvM, JvF, MZ, AR and PB reviewed and edited the manuscript. IvM, JvF, and AR supervised the project.

## Competing interests

The authors declare that they have no conflict of interest. Some authors are members of the editorial board of journal PNAS.

## Acknowledgements

We sincerely thank Marius Luder and Nina Nagel for their help in equipment installation and maintenance, sample collection and preparation and data processing. We thank Barbara Strobl, Joshua Haas, Alex Karapanscev, Rick Assendelft, Florian Käslin, Pascal Arpagaus, Florian Lustenberger, Kari Steiner, and Stefan Boss for helping with fieldwork in the Alptal catchment. This work was funded by the Swiss National Science Foundation grant (PR00P2_185931) awarded to JvF.

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
