# Peer review of "Changes in the flowing drainage network and stream chemistry during rainfall events for two pre-Alpine catchments"

_Hydrology and Earth System Sciences, 2024_

## Author Response (AR1)

**Response to reviews and to the Associate Editor's assessment**

Hydrology and Earth System Sciences Manuscript ID hess-2024-67 "Changes in flowing drainage network and stream chemistry during rainfall events for two pre-Alpine catchments" by Dr. Bujak-Ozga et al.

We would like to express their gratitude for the valuable feedback received. All comments on the manuscript have been instrumental in our revision and improved the manuscript. Below is a point-by-point response to all review comments with details of how we have used them to modify our manuscript.

Note: Reviewer comments are printed in *italics*. Line numbers in *italics* in the reviewers' comments refer to the originally submitted version of the manuscript. Line numbers in our responses refer to the revised version of the manuscript without track changes. We also provide a version of the manuscript with all the changes tracked as a separate file.

**Reviewer 1 Comments and Responses**

**General comments**

This manuscript presents a very interesting study, linking dynamics of expansion and contraction of intermittent stream networks in a pre-alpine setting to resulting streamflow chemistry and streamflow generation mechanisms during several rainfall events. The hydrochemistry and hydrometric datasets are impressive in their extent and temporal resolution.

Datasets like this are rare, and they allow addressing the relevance of hydrologic connectivity (either between landscape elements and streams or along the stream) for the patterns and dynamics of the export of solutes. This is a very important question where we still have a lack of understanding.

In general, I think, this study is a valuable contribution to this field of research. It is well written and describes clearly the research objectives, the study sites and experiments, and the results. I only have a few specific comments. What I think is problematic is that results and discussion are presented together and not separately. There is this constant mixing of description of results and explanations and interpretations, and the text keeps jumping between the different aspects of the study. In my opinion, the focus on the answers to the (important!) research questions posed at the end of the introduction gets lost by this setup. I would strongly recommend splitting results and discussion. After describing the results in a concise way with the very informative figures the authors used, they can come back to the research questions in the discussion and answer them. At the moment, the reader has to look for the answers to the research questions in the various parts of the results and discussion sections. I think that the authors could elaborate more on their second research question, i.e., the comparison between the two topographically different catchments. Is the main impact of topography the influence on the flow network and the "state of connectedness"? Also, the link of the research findings particularly to intermittent streams is missing in the interpretation. At the end of the discussion, the conceptual model of the flowing drainage network and solute export could be presented as a summary of the new understanding that was gained, instead of mentioning it in the conclusions only. By presenting results and discussion separately, the main findings of this very interesting study and the interpretations and implications could be presented in a more focused and clearer way.

**Dear Reviewer,**

Thank you for your thoughtful and comprehensive feedback on our manuscript. We are

pleased that you found our study valuable and well-written and appreciate your recognition of the importance and uniqueness of the datasets we presented. We acknowledge the need to improve the structure and clarity of our manuscript, particularly regarding the presentation of results and discussion. Below, we address your general and specific comments in more detail.

**General comments**

1. Separation of Results and Discussion: We understand your concern about the mixing of results and explanations and interpretations. While we appreciate the suggestion to separate results and discussion (and would in general also choose this structure), we believe that in this manuscript integrating them allows for a more cohesive narrative, where results can be immediately interpreted and contextualized. Because very different results are presented in this manuscript, we consider this beneficial for this case. For example, we think that it is necessary to discuss the hydrochemical characteristics of the water sources and describe how these results are similar to those of previous studies in the area, before we continue to use these data for other (e.g., mixing) analyses. To address your concern, we reorganized the structure of the paper to more clearly distinguish the presentation of data from their interpretation within each section. We think that this helps to maintain a clear focus on the research questions without separating the sections entirely.

2. Focus on Research Questions: We revised the manuscript to ensure that the discussion consistently revisits and directly answers the research questions posed at the end of the introduction. We highlighted the answers to these questions more clearly to ensure they are easily identifiable to the reader. Specific changes include:

- We changed the title of section 3.2. to resemble more the research question it answers (Now it is: 'How do the flowing drainage networks and stream chemistry change during events').

- We added the phrase "(...) how the flowing drainage network and solute concentrations change during small events (...)" (L410) in the section 3.2.2. to explicitly point out that the text addresses these points (Question 1).

- We edited texts in sections 3.3.5., and 3.4.5., to highlight the differences between the two catchments and strengthen the link to research question 2. We slightly reworded our research question 2, to make it clearer. Now it is: "How do solute concentrations and flowing drainage network dynamics differ for two catchments with different geomorphic channel networks?"

- We added section 3.5. (L716-736) which specifically summarizes the findings focusing on addressing research questions.

In addition, we have made some minor changes in the wording of some other sections, including the abstract and conclusions to make the focus even clearer.

3. Comparison Between Catchments: We elaborated more on our second research question concerning the comparison between the two topographically different catchments. Specifically, we discuss how topography influences the stream network and the state of connectedness in more detail (entire new section 3.5. L716-736). Moreover, a clearer presentation of the conceptual model of the flowing drainage network and solute export (edited texts in sections 3.3.5., and 3.4.5.) enabled us to better illustrate the similarities and differences between these catchments, thereby addressing our research questions more effectively.

4. Link to Intermittent Streams: We strengthened the link of our research findings to intermittent streams in the conclusions section. This highlighted the significance of our study in the context of intermittent stream networks and their unique characteristics. We added (L770-771): "Overall, this study contributes to the growing body of scientific literature on

intermittent streams and the associated dynamics in stream chemistry (e.g., Hale and Godsey, 2019; Zimmer and McGlynn, 2018; von Schiller et al., 2017; Merbt et al., 2016)."

5. Conceptual Model: We presented the conceptual model of the flowing drainage network and solute export first separately for the two catchments (in the thoroughly edited sections 3.3.5., and 3.4.5.), and then jointly, as a summary at the end of the discussion (new section 3.5). This provided a clear and concise synthesis of the new understanding gained from our study before the conclusions section. We used these sections to highlight the answers to the research questions again as well.

**Detailed comments**

L 147: I believe you mean the METER Group company

We corrected this to "METER Group."

L 176: "trickling" instead of "tricking"

We changed "tricking" to "trickling."

L 228: Start new sentence after E2

We started a new sentence after "E2."

Section 2.5.2: If I understand correctly, the method of creating maps of flowing drainage networks and the indices does not allow a statement on connectedness or connectivity. If 50% of the stream reaches were classified as flowing, this could still mean a quite fragmented state (e.g. if every second stream reach was flowing). Maybe the authors could comment on this.

When we stated that we create maps of flowing drainage networks, we aimed to convey that we record the flowing or non-flowing state of each reach at every time step and maintain spatial information about their locations and connections (derived from mapping surveys and stored in our database). Using these data, we visually assessed the connectivity of the stream network on the surface. Specifically, we generated time-lapse maps showing flowing and non-flowing reaches in different colors.

An alternative approach would be to calculate an index expressing network connectivity (e.g., length of connected network), but we chose the more qualitative first method to identify which parts of the catchment re-wet and dry up first. This method was also faster to program based on our data storage format.

This section focuses solely on surface connectivity. We discuss the potential use of our hydrochemical data to understand subsurface connectivity in other sections. We clarified these points in the relevant sections. Specifically, in section 2.5.2 we wrote (L243-246): "This is repeated for each reach, leading to a continuous time series of "flow" and "no flow" for each stream reach and spatial maps of the flow/no-flow conditions for each channel reach at a 10-min resolution. We used these spatial maps, to visually assess the connectivity of the flowing stream network.". Moreover, in section 2.5.4. (L281-283) we added: "Both isotope-based hydrograph separation and EMMA were used to better understand the contributions of different flow pathways to streamflow and thus surface and subsurface connectivity."

*Caption Fig. 4: I think there was a mix-up of how the panels a-f were described in the caption. Please check. E.g. "time-series of hydrologic variables…" are panels b and e and not b and c.*

We revised the caption of Figure 4 to accurately describe the panels. We apologize and appreciate you pointing out this mix-up.

L 473: redundant "from areas"

We removed the redundant "from areas."

L 689: Fig. 6! Not Fig. 8.

We corrected the references in conclusions to Fig. 6 and Fig. 4 instead of Fig. 8.

We think that these revisions have significantly improved the manuscript, making our findings clearer and more accessible to readers. Thank you again for your valuable feedback.

**Reviewer 2 Comments and Responses**

**General comments**

In this manuscript the authors explore the relation between stream network extent and outlet stream chemistry during precipitation events. I really enjoyed reading the introduction, it serves as a nice overview of existing work and the importance of understanding how intermittency impacts chemistry, and highlights that it is relatively understudied. The posed research questions are important, and well stated. There has been extensive work on event water chemistry, but limited study of chemistry in intermittent streams, which makes this dataset unique.

I have one major comment regarding research questions in the results and discussion section. I appreciate that the sections are merged as there is lots of quantitative geochemistry in this paper and it is helpful to have discussion immediately following results to provide context for conclusions. However, the answers to the research questions are lost in the latter portion of the manuscript. Some reorganization of the results/discussion to clearly highlight the answers to the research questions would improve the manuscript, specifically in sections 3.2 and 3.3. One suggestion is to group analysis by solute type (i.e. weathering derived, nutrient, metal), since they seem to behave similarly, rather than jumping from one solute type to the next in each section. Another suggestion is to increase discussion of the conceptual figure, as detailed below.

The conceptual figure is an effective summary of the paper, but 1) I am confused by some of the flowpaths and colors, and 2) suggest including more text interpretation of the figure to tie it back to the research questions. Specifically, do the letters in the block diagrams correspond to color at all? The purple color is labeled groundwater flow, so I am not quite sure what the dashed purple ellipses are around the stream. Do the red stream lines represent a dry channel? Why then is panel a, spot A called streamflow if it is red, but spot E around a red stream in panel c is no flow? What does the light blue color in the subsurface of panel D represent? Similarly, why are some lines dashed and others soild? A detailed legend figure would help interpretation of this figure. It would also strengthen the paper to add a summary section discussing this conceptual figure in the context of the research questions. Much of the important conclusions are found at the end of paragraphs, and so discussion of them together would help.

Thank you for your thorough and insightful review of our manuscript. We were pleased to hear that you enjoyed the introduction and found our research questions both important and well-stated. Your recognition of the uniqueness of our dataset in the context of intermittent stream chemistry is highly appreciated as well. We have carefully considered your comments and suggestions to improve our manuscript and found them very useful to strengthen our manuscript. Below, we provide detailed responses to each of your points.

Reorganization of Results/Discussion: We agree that reorganizing sections 3.2 and 3.3 to better highlight the answers to our research questions will enhance the manuscript. While we do not agree that it is useful to entirely rewrite these sections to group solutes by type (as we already do this to some extent). Still, we reorganized the content to improve clarity and flow and, as suggested, added more references to the Fig. 6 (conceptual model, especially in sections 3.3.5., and 3.4.5.). Moreover, as also described in our response to the other reviewer, we revised the Results and discussion section to ensure that the discussion consistently revisits and directly answers the research questions posed at the end of the introduction. These changes should increase clarity and help readers more easily follow the answers to the research questions.

Conceptual Figure Clarification: As also requested by the other reviewer, we presented the conceptual model of the flowing drainage network and solute export (in the thoroughly edited sections 3.3.5., and 3.4.5.) and then, as a summary at the end of the discussion (Section 3.5. L716-736). This provided a clear and concise synthesis of the new understanding gained from our study before the conclusions section. Moreover, we revised the conceptual figure to enhance clarity and added a detailed figure legend and extended the caption explaining the colors, flowpaths, and line types. Specific changes include:

- We used a visual legend instead of letters and provided detailed explanations for the dashed versus solid lines (figure caption and in the added legend) and the significance of each color in the figure.
- We added more text to the figure caption to make it clearer and to tie the figure back to our research questions (i.e., 'Figure 6: Conceptual diagram showing the flowing drainage network in Lan and Cha and the main processes leading to streamflow during baseflow conditions (a, c) and rainfall events (b, d). The related temporal variations in solute concentrations at the catchment outlet are shown for nitrate (e), iron (f), calcium (as an example of weathering-derived solutes) (g), and potassium and chloride (h). The green and orange lines in e-h represent the solute dynamics for Lan and Cha, respectively. Solid lines on the surface in a-d represent channels without flow (red), channels that were already flowing before the start of the event (dark blue), and channels that started to flow during event (light blue). The dashed lines represent source areas that contributed to streamflow before event (dark blue; groundwater), and during events (light blue; groundwater, saturated overland flow, and interflow through the topsoil). The two green lines for the nitrate dynamics in e represent the different magnitude of response in Lan for events with dry ("flush") and wet antecedent conditions. In Lan (with flatter topography and longer channel network), the flowing drainage network expands rapidly during rainfall. This leads to a rapid increase in connectivity of the flowing drainage network, which shapes the solutes responses, especially for weathering derived solutes. Moreover, re-emergence of flow in previously dry channels leads to an increase in nitrate concentrations at the outlet ("flush") that depends on the antecedent conditions. In contrast, the more stable flowing drainage network in the steeper Cha results in more gradual changes in solute concentrations and no nitrate flush. The connection of saturated areas to the channel network facilitates the transport of iron from these saturated areas.')'.

- We also included summary sections (3.5., 3.3.5., and 3.4.5.) expanding discussion of the conceptual figure in the context of our findings.
- We, moreover, made some other minor edits to the figure, to make it clearer for the reader.

**Detailed comments:**

60-61 – Please provide citation for this statement.

We added the appropriate citation to support the statement, i.e., Brinkerhoff et al. (Science, 2024), Alexander et al. (JAWRA, 2007). We also simplified the sentence: "The dynamic variations in flow conditions along drainage networks can influence the quantity and quality of streamwater in downstream reaches (Brinkerhoff et al., 2024; Alexander et al., 2007)."

66 – Correct to "Warix et al. (2023) used CFC-12 & 3H"

We corrected the reference to "Warix et al. (2023) used CFC-12 & 3H.

209 – Are you reporting solute totals? Please specify.

We have clarified that we are reporting solute total mass in this section. Specifically, we added (L212): "Solute concentrations are reported as the full solute in mg  $l^{-1}$ , not as the elemental component."

309-323 - I suggest adding some mineral saturation indices to support the conclusions in this paragraph. The hypothesis that baseflow is saturated with respect to secondary calcium products seems reasonable but speculative and could be made more concrete by calculating the saturation index of calcite for groundwater samples.

We agree that calculating the saturation index of calcite for groundwater samples would strengthen our hypothesis. Unfortunately, we cannot calculate this index because we did not measure the pH of the groundwater. Nonetheless, as noted in the review comment, the hypothesis that differences in measured calcium concentrations between baseflow and groundwater in Lan are due to varying conditions and the saturation of secondary calcium products seems reasonable. This hypothesis is supported by values reported in other studies conducted near the Lan area and Alptal sub-catchments. Hagedorn et al. (Geoderma, 2001) reported acidic pH values (pH 5.1-5.9) for deeper soils (depths >30 cm) in the Lan area, while Fischer et al. (Hydrological Processes, 2015) reported basic pH values (median >8.0) in the baseflow of the Alptal sub-catchments.

314 – Do you have pH observations to support Lan being more acidic than Cha?

We did not measure pH, however, it is common that forest soils are more acidic. We provide references to more studies, e.g., Huang et al. (Forest Ecology and Management, 2022) and Berthrong et al. (Ecological applications, 2009) to support this claim.

**473 – "from areas" is repeated**

We removed the repeated phrase "from areas".

**469-477 - I had to read these lines several times to understand the proposed flow mechanisms. Can some of these sentences be merged or shortened for conciseness?**

We revised these lines for conciseness and clarity. We think that these changes have improved the readability and understanding of the proposed flow mechanisms. Now it is (L483-491): 'This suggests that the connection of the flowing drainage network caused an increase in the contribution of event water at the catchment outlet. However, direct precipitation on the flowing stream channel was much less than the calculated event water flux (Fig. S13-S16). Furthermore, event water fractions remained high even after rainfall ceased and discharge decreased (Fig.4, Fig.S3-S8). This indicates that event water must have originated from areas outside the channel network, such as saturated overland flow and shallow (root zone) interflow. Saturated overland flow has been observed across many locations in the neighboring Studibach catchment, particularly in the wet meadow areas (van Meerveld et al., 2019). Infiltration excess overland flow likely plays a minor role as the soil surface hydraulic conductivity typically exceeds the rainfall intensities (van Meerveld et al., 2019; Wadman 2023).'

510 – This idea could use some clarification. Where is the decrease in concentrations happening? If GW solute concentrations are higher, and the groundwater table elevation rises how does that lead to a decrease in concentration? I assume mixing with dilute soil waters, but this should be made explicit.

We intended to describe a decrease in concentrations at the catchment outlet as a result of stratification in groundwater concentrations and the rising groundwater table (process described in e.g., Knapp et al. (Hydrol. Process., 2022)). We recognize that the original wording may have been confusing and we revised the paragraph for clarity. Now it is (Lines 521-531): "Previous research has suggested that vertical concentration profiles of weatheringderived solutes can cause dilution patterns in streamwater as well (Botter et al., 2020; Knapp et al., 2022; Seibert et al., 2009). In Lan, calcium concentrations in the shallow soils increase with depth below the surface (Hagedorn et al., 2001). This was also observed in the neighboring Studibach catchment (Kiewiet et al., 2020; Bruppacher, 2022). Vertical stratification of weathering-derived solute concentrations in soil and groundwater could potentially lead to a decrease in concentrations at the outlet as the groundwater levels increase during rainfall events and water with lower concentrations in shallower soil layers is transported and becomes hydrologically connected to the stream. However, this process likely plays a minor role in Lan because the observed concentrations for weathering-derived solutes (calcium, sulfate, sodium, magnesium, silica) correspond well with those expected from a simple mixing with event water (e.g., Fig. 4). This suggests that the primary driver for these changes in solute concentrations is the dilution of the groundwater with event water, rather than contributions from different parts of the vertical soil profile."

Figure 3 -The y-limits should be adjusted to the max observed data. Some plots (e.g. Fe) have lots of white space.

We adjusted the y-limits in Figure 3 to reduce white space.

We believe these revisions enhanced the clarity and impact of our manuscript. Thank you again for your valuable feedback.

Kind regards, Izabela Bujak-Ozga on behalf of all co-authors

---

## Author Response (AR2)

**Response to reviews and to the Associate Editor's assessment**

Hydrology and Earth System Sciences Manuscript ID hess-2024-67 "Changes in the flowing drainage network and stream chemistry during rainfall events for two pre-Alpine catchments" by Dr. Bujak-Ozga et al.

We appreciate the detailed feedback provided. The comments on the manuscript have been helpful in enhancing its quality. Below, we present a detailed point-by-point response to the review comments, explaining how they were incorporated into the revised manuscript.

Note: Reviewer comments are printed in *italics*. Line numbers in italics in the reviewers' comments refer to the previously submitted version of the manuscript. Line numbers in our responses refer to the revised version of the manuscript without track changes. We also provide a version of the manuscript with all the changes tracked as a separate file.

**Reviewer 2 Comments and Responses**

Line 684: I am slightly confused about the use of runoff in this sentence, do you mean shallow source?

Thank you for your comment. You are correct that the term "runoff pathways" could be confusing in this context. Our intent was to emphasize shallow flow pathways as the likely source of chloride and potassium, rather than a connectivity to a different groundwater source. To address this, we have revised the text and replaced the original sentence ("This quick response suggests that a contribution from shallow flow pathways is more likely runoff pathways for chloride and potassium than connectivity to a different groundwater source.") with "This quick response suggests that a contribution from shallow flow pathways is a more likely source of chloride and potassium than connectivity to a different groundwater source." (L683-684).

Line 759: Do you mean that event flow is due to the direct contribution of rainfall in the channel, and only at the start of events? Based on the discussion, I thought the rewetting was due to the connection of saturated areas, not limited precipitation directly over the channel. A slight rewording will help here.

We edited the manuscript text to make the mechanisms contributing to event water clearer. Indeed, direct rainfall onto the channel plays an important role only at the very beginning of events. As events progress, other sources (e.g., overland flow from saturated areas or quick interflow through the topsoil) become more important sources of event water.

We have revised the text to address this and clarify the distinction between these mechanisms. Before corrections the text was: "The event water contributions could be explained by rain falling on the channels only at the beginning of rainfall events. During wetter conditions, overland flow from saturated areas (Fig. 6) or quick interflow through the topsoil must have contributed event water to the stream as well.". After corrections it is: "At the beginning of rainfall events, event water contributions could be explained by rain falling onto the channels. However, as the events progress, this is insufficient to account for the observed event water flux. Thus, additional contributions from overland flow from saturated areas (Fig. 6) or quick interflow through the topsoil are also important, particularly under wetter conditions when saturated areas become connected to the stream." (L759-763)

Figure 6: Perhaps it is my screen but the "green" lines in 6e-h appear blue not green.

Thank you for pointing this out. Indeed, it should be blue. We corrected it.

Besides addressing the requested changes, we have also proofread the manuscript again and made two minor corrections: (1) corrected a typo in the equipment name on line 210, (2) updated the reference to the dataset on lines 780 and 849, and (3) corrected a typo in the abstract on line 14.

We think that the careful reading of the reviewer and suggested revisions have further improved the manuscript by avoiding confusing and making our findings clearer and more accessible to readers. Thank you again for your valuable feedback.

Kind regards, Izabela Bujak-Ozga on behalf of all co-authors